# Tabby: A Language Model Architecture for Tabular and Structured Data Synthesis

**Sonia Cromp**                                                      *sonic@cs.wisc.edu*
*Department of Computer Sciences*
*University of Wisconsin-Madison*

**Satya Sai Srinath Namburi GNVV**                    *satya.namburi@gehealthcare.com*
*GE HealthCare*

**Catherine Cao**                                                 *ccao@netflix.com*
*Netflix*

**Mohammed Alkhudhayri**                                  *malkhudhayri@wisc.edu*
*Department of Computer Sciences*
*University of Wisconsin-Madison*

**Samuel Guo**                                                   *sguo258@wisc.edu*
*Department of Computer Sciences*
*University of Wisconsin-Madison*

**Nicholas Roberts**                                            *nick11roberts@cs.wisc.edu*
*Department of Computer Sciences*
*University of Wisconsin-Madison*

**Frederic Sala**                                                 *fredsala@cs.wisc.edu*
*Department of Computer Sciences*
*University of Wisconsin-Madison*

**Reviewed on OpenReview:** *https://openreview.net/forum?id=b9FPVnb0Bn*

## Abstract

Large language models (LLMs) have greatly improved the quality of synthetic text data. We aim to extend these advances to *tabular* data with **Tabby**, a simple but powerful post-training modification to the standard Transformer language model architecture, enabling its use for tabular dataset synthesis. Tabby represents differences across columns using Gated Mixture-of-Experts, with column-specific sets of parameters. Empirically, Tabby results in data quality near or equal to that of real data. Pairing Tabby with **Plain**, our novel tabular training technique, we observe up to a 7% improvement in quality (measured by MLE) over previous methods. Additionally, our approach is *more flexible* than prior strategies and extends beyond tables, to more general structured data. In a structured JSON setting, Tabby outperforms all other methods by 2-3 points and is the only approach with MLE equal to the upper bound of non-synthetic data.

## 1 Introduction

Modern life is built on tabular data: airplane black boxes, website analytics and hospital patient records are just a few examples of this versatile modality. Despite widespread use of tables and repeated calls for improved table modeling approaches (Fang et al., 2024; Davila et al., 2024), the tabular modality has received less attention in recent deep learning research than images or text (van Breugel & van der Schaar, 2024).

Progress towards realistic tabular data synthesis encounters several key challenges. *First*, table columns often exhibit complex interdependencies. *Second*, many tabular datasets mix multiple datatypes. A single table

might contain free-text fields, numerical features, and even nested JSON or dictionary columns. *Third*, although the order of tokens within one column is important, the order of columns with respect to each other is usually not meaningful and is a potential source of spurious correlations during training. How best to design model architectures and training techniques that address these issues is an open question.

There have been notable efforts to adapt several model architectures to tabular data, recently focusing on generative adversarial networks (GANs) (Xu et al., 2019), LLMs (Borisov et al., 2022) and diffusion models (Kotelnikov et al., 2022). However, because these architectures were each designed with images or text in mind, significant preprocessing must be made to tabular datasets in order to allow their use.

For these reasons, works including van Breugel & van der Schaar (2024) have called for the development of pretrained *Large Tabular Models (LTMs)* to fill a similar role to text and image foundation models, such as GPT (Achiam et al., 2023) or Stable Diffusion (Blattmann et al., 2023). Unfortunately, the creation of an LTM requires (1) large, diverse tabular pretraining sets which have not yet been cu-

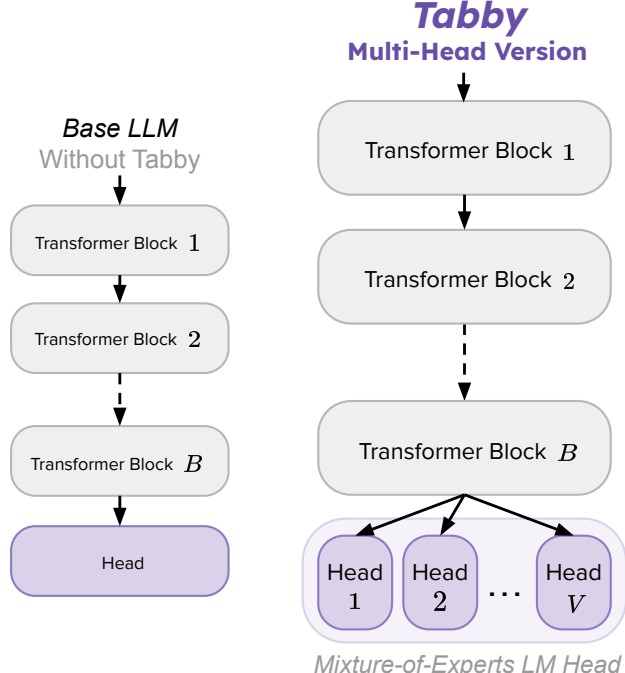

Figure 1: Tabby Multi-Head modifications (right) compared to an original, Non-Tabby LLM on left.

rated, (2) a specialized tabular model architecture which has yet to be designed, and (3) substantial compute resources for pretraining. These challenges are even more pronounced in the development of foundation models for structured *non-tabular* modalities, such as JSON and geospatial data.

This work takes an initial step towards LTMs with **Tabby**, **a post-training modification to the standard transformer LLM architecture to enable tabular and other structured data synthesis**. After training on text data—but before finetuning on structured data—Tabby replaces select LLM blocks with deterministic *Mixture-of-Experts (MoE) layers* (Shazeer et al., 2017), allowing each tabular column, JSON attribute or other structured feature to be modeled by a dedicated set of parameters in the LLM. The greater expressivity afforded by this change results in higher-fidelity synthetic data. Fine-tuning with our novel *Plain* technique results in still higher performance. We show that even small Tabby models are capable of outstripping large non-Tabby LMs with parameter counts orders of magnitude greater.

To our knowledge, Tabby is *the first architecture modification to make LLMs better-suited to table generation*. Using a pretrained LLM as a starting point allows Tabby to take advantage of its text pretraining, avoiding the logistical challenges of training a LTM fully from scratch. We find that, according to standard metrics, **Tabby produces synthetic data near- or at-parity with real data on 5 out of 8 table datasets**. Additionally, Tabby is not limited to tables and can be easily extended to other structured data. We validate this by **synthesizing nested JSON data at-parity with real data** as well. Our contributions are:

- We introduce *Tabby*, [1] a simple architecture modification that allows transformer-based LLMs to synthesize more realistic tabular and structured non-tabular data.

- We demonstrate that Tabby produces higher-quality synthetic data for 4 out of 6 popular tabular datasets, is compatible with more tabular data than prior high-quality synthesis methods, and can be extended beyond tables to a broader class of *structured data modalities*.

- We introduce our novel tabular LLM training method, *Plain*. Named after its surprising simplicity compared to prior table training approaches for LLMs, Plain increases data quality in all 8 datasets when used together with or independently of Tabby, compared to the prior SOTA LLM approach.

---

[1]Codebase: https://github.com/soCromp/tabby

## 2 Tabby Architecture & Plain Train Method

We now formally introduce our two novel contributions for LLM table synthesis: *Tabby* is an architecture modification that may be applied to any transformer-based language model (LM) (Vaswani, 2017), and *Plain* is a training technique for training any (Tabby or non-Tabby) LM on tabular data. Tabby and Plain are especially powerful together: Tabby increases model expressivity in a way that is specially-suited to tabular data, while Plain allows the model to more effectively fit to the key features of a tabular dataset during training, yielding more realistic data synthesis.

In Section 2.1, we describe Tabby for tabular or other structured data. In Section 2.2 , we outline the process for training an LM on tabular data using our Plain training technique. Then, in Section 2.3, we provide additional insight into how Tabby models are trained (using Plain, or pre-existing LLM table training techniques such as GReaT (Borisov et al., 2022)) by comparing the training process's forward pass and loss calculation for a Tabby model with a non-Tabby model.

### 2.1 Architecture of Tabby Models

The intuition behind the Tabby modification is simple: we want to allow the model to learn individual columns as distinct–but interdependent–tasks. *The right side of Figure 1 depicts our best-performing Tabby model for tabular data*, where Tabby modifies only the language modeling head.

To provide a general definition of a Tabby model, consider a tabular dataset with $V$ columns. Let the order of blocks within an arbitrary transformer-based LM be represented as $[L_1, L_2, \ldots, L_H]$. We apply the MoE technique by replacing an LM block $L_a$ with a vector $\Lambda_a = [L_{a,1}, L_{a,2}, \ldots, L_{a,V}]$ of $V$ blocks. Thus, a Tabby model with one MoE block $\Lambda_a$ is represented

$$[L_1, L_2, \ldots, L_{a-1}, [L_{a,1}, L_{a,2}, \ldots, L_{a,V}], L_{a+1}, \ldots, L_H].$$

The dataset's $i$-th column is modeled by $L_{a,i}$ within $\Lambda_a$. In this way, Tabby's MOE layers use a deterministic, schema-based routing function where the chosen expert is determined by the current column index. The gating function can be expressed mathematically as $y_i = \sum_{j=1}^{V} \mathbb{1}\{i = j\} f_j(x_i)$, where $x_i$ denotes a position within the $i$-th column, and $f_j$ is the $j$-th expert.

This technique may be applied to any set of layers within the model. While we focus on the language modeling (LM) head [2] in Figure 1 and Section 3 evaluations, we also conduct experiments applying Tabby to the transformer multi-layer perceptrons and attention blocks in Appendix E. We refer to Tabby models with MoE LM Heads as *Tabby Multi-Head (MH)* models.

### 2.2 The Plain Technique for Fine-Tuning LLMs on Tabular Data

Suppose our trainset contains $N$ rows and column names denoted by $v_1, v_2, \ldots, v_V$, such that $v_i^j$ represents the value of the $j$-th row in the $i$-th column. To provide the LM with its expected text modality input, we convert the $j$-th row as follows, where `<EOS>` is the end-of-sequence token and `<EOC>` is a specialized end-of-column token which we introduce to divide the text between columns:

"`<BOS>` $v_1$ is $v_1^j$ `<EOC>` $v_2$ is $v_2^j$ `<EOC>` $\cdots$ $v_V$ is $v_V^j$ `<EOS>`"

Converting the tabular dataset in this fashion allows an LM to fine-tune on the dataset in a normal sequence-to-sequence style. Because Plain encodes data the same way as prior LLM table training techniques, GReaT and Tabula (Borisov et al., 2022; Zhao et al., 2023), Plain does not require more FLOPs than prior methods.

During inference, the prompt for each row is the beginning-of-sequence token `<BOS>`. During generation, the LM will output text in a similar format to the training data, which can then be parsed into tabular data as desired. We note that *the simplicity of Plain is particularly impressive given its favorable performance compared to prior LLM table training methods, as we show in Section 3.*

### 2.3 Tabby Training

Now that we have introduced Tabby and the Plain training method, we are able to provide further insight into aspects of the training process unique to Tabby. Suppose that we construct a Tabby model from a base

---

[2]"LM head" refers here to the language model output layer, distinct from attention heads in the MLP blocks.

LM by replacing one of its blocks $L_a$ with an MoE set $\Lambda_a$. At the beginning of fine-tuning the Tabby model, weights for each block in $\Lambda_a$ are initialized to equal the weights of $L_a$. These column experts then diverge from each other over the course of fine-tuning.

The Tabby training process requires only slight modifications compared to other LMs for tabular data. Instead of representing each training row as one string, we convert each row into a list of $V$ strings:

$$["v_1 \texttt{ is } v_1^j \texttt{ <EOC>", "} v_2 \texttt{ is } v_2^j \texttt{ <EOC>", } \cdots, "v_V \texttt{ is } v_V^j \texttt{ <EOS>"}]$$

Internally, the Tabby model begins by training on column 1 with prompt <BOS>, attending to tokens 0 through $k-1$ when predicting the $k$-th token. After computing the loss on column 1, this column's tokens are appended to the prompt used to train column 2. Backpropagation is performed at the end of a batch of rows. The prompt when training on column $i$ is

$$"\texttt{<BOS>}v_1 \texttt{ is } v_1^j \texttt{ <EOC> } v_2 \texttt{ is } v_2^j \texttt{ <EOC> } \cdots v_{i-1} \texttt{ is } v_{i-1}^j \texttt{ <EOS>}"$$

Because we calculate losses for each column separately, we are able to monitor the performance of each column individually during training. This favorable side-effect is demonstrated in Section 3.4.

## 2.4 Extensions

We address two additional aspects of Tabby and Plain: (1) generalizations that go beyond tabular data and (2) optimizations for datasets with large numbers of columns.

**Synthesis for general structured modalities:** The flexibility in Tabby MoE layer design enables extensions to a variety of structured datatypes, such as *hierarchical* data. For example, we create a model for nested JSON data by applying Tabby recursively in Figure 5. The JSON structure is preserved inherently in the model, so that Plain's method of representing data features does not need to be modified to indicate nested features. As we show in Section 3.4, the combination of Plain and Tabby is the only synthesis approach to reach equal performance to real, non-synthetic data.

**High-dimensional data:** Because Tabby MoEs contain one block per dataset column, model parameter count is proportionate to the number of data features. In practice, however, techniques such as parameter sharing (Ravanbakhsh et al., 2017) can drastically reduce the number of parameters to represent a Tabby model. Additionally, Tabby may be implemented so that only one block in the MOE layer is in memory at a time, a potential strategy to achieve memory requirements identical to a non-Tabby model.

## 3 Experimental Results

Our evaluations seek to assess the following claims:

**Claim 1**: Plain-trained Tabby models generate higher-quality tabular data than prior approaches.
**Claim 2**: The Tabby architecture modification allows smaller LLMs to achieve similar or better synthetic data fidelity than LLMs with higher parameter counts.
**Claim 3**: Tabby architecture modifications may also be applied to other structured data beyond tabular data, resulting in higher-quality synthetic data for these modalities as well.
**Claim 4**: Tabby's loss formulation allows for convenient tracking of per-column performance at training time, leading to better understanding of model behavior.
**Claim 5:** Tabby's language modeling capabilities enable it to capture the underlying semantic structure of column values unseen during pretraining, allowing it to generate novel yet realistic values beyond the pretraining distribution.

After providing key evaluation setup details in Section 3.0, we compare Tabby to a broad array of prior works on diverse tabular datasets in Section 3.1 to evaluate Claim 1. As Tabby may be applied to any transformer-based LM, we explore Claim 2 for LMs of varying sizes in Section 3.2. To demonstrate Claim 3, we apply Tabby to a nested (JSON) dataset in Section 3.3. In Section 3.4, we investigate how Tabby adapts to individual columns within a dataset during finetuning as a demonstration of Claim 4. Lastly, we demonstrate Claim 5 in Appendix E.5

Table 1: Summary statistics of datasets. The first three columns list the number of rows in each data split, while the next two columns display the number of categorical versus numerical features, respectively. The rightmost column details whether the dataset is considered a classification (C) or regression (R) task in downstream evaluations.

|  | N Train | N Validation | N Test | # Cat. | # Num. | Task |
|---|---|---|---|---|---|---|
| **Diabetes** (Kahn, 1994) | 576 | 57 | 135 | 0 | 8 | C |
| **Travel** (Tejashvi, 2023) | 715 | 71 | 168 | 4 | 2 | C |
| **Adult** (Becker & Kohavi, 1996) | 36631 | 3663 | 8548 | 8 | 6 | C |
| **Magic** (Bock, 2004) | 17117 | 1711 | 1902 | 0 | 10 | C |
| **Shoppers** (Sakar & Kastro, 2018) | 11097 | 1109 | 1233 | 7 | 10 | C |
| **Abalone** (Nash et al., 1994) | 3132 | 313 | 732 | 1 | 7 | R |
| **Rainfall** (Zaman, 2018) | 12566 | 1256 | 2933 | 2 | 1 | R |
| **House** (Pace & Barry, 1997) | 15480 | 1548 | 3612 | 0 | 8 | R |

### 3.0  Setup

We detail here our experiments' essential information, including baselines, evaluation datasets and metrics. Additional details are located in Appendix D.

**Baselines and Comparisons:** We evaluate a variety of recent tabular synthesis techniques.

*LLM Approaches:* Prior LLM table synthesis approaches are limited to the development of training techniques. We compare Tabby and Non-Tabby LLMs trained under three different paradigms:

1. Our lightweight and simple ***Plain*** training paradigm, detailed in Section 2.2.

2. **GReaT** (Borisov et al., 2022), which encodes tabular data similarly to Plain, but permutes the orders in which columns are presented in training and imposes some conditional restrictions at sample time. For more details, see Section 4.

3. GReaT combined with TapTap (Zhang et al., 2023) and Tabula (Zhao et al., 2023). We abbreviate this combination as **GTT**. TapTap pretrains the LLM on tabular data, while Tabula encodes each categorical column into an ordinal format by replacing each unique column value with an integer.

To align with the prior works (Borisov et al., 2022; Zhang et al., 2023; Zhao et al., 2023), LLM methods use Distilled-GPT2 (DGPT2) (Radford et al., 2019) as a base model unless otherwise stated.

*Non-LLM Approaches:* To represent non-LLM tabular synthesis techniques, we include CTGAN+ (Zhao et al., 2022), CTGAN (Xu et al., 2019) and TVAE (Xu et al., 2019), the leading GAN and VAE approaches, as well as diffusion models TabSyn (Zhang et al., 2024), TabDiff (Shi et al., 2025), Tab-DDPM (Kotelnikov et al., 2022) and Forest Diffusion (Jolicoeur-Martineau et al., 2024). Although diffusion models are a SOTA approach to achieving high MLE scores, they do so under strong assumptions and are incompatible with many tabular datasets—see Figure 4 and Section 4.

Additional details on how models are trained and sampled are available in Appendix D.

**Datasets:**  We evaluate Tabby on eight common tabular datasets, which are summarized in Table 1. The majority of these datasets are standard for the evaluation of tabular synthesis techniques, allowing for easy comparison with prior approaches. For more information on these datasets, see Appendix B.

**Metrics:**  We focus on ***machine learning efficacy (MLE)*** **(Dankar et al., 2022), the standard metric for quantitative evaluation of synthetic tabular data**. In brief, MLE compares the performance of downstream classifiers that were trained using either real or synthetic data.

Our MLE results in the following sections are interpreted as follows: the downstream classifier that is trained using non-synthetic, real data is considered the upper bound and **any MLE score higher than this "Non-Synthetic" classifier's score is considered the best. If no score surpasses the "Non-Synthetic" score, then any highest score is considered the best**. Figure 2 summarizes the MLE calculation

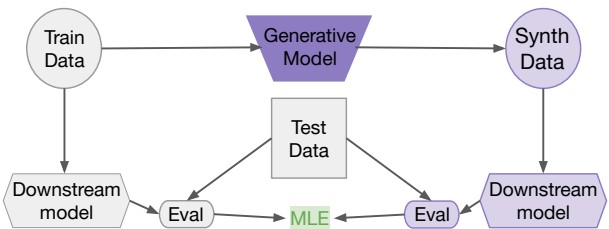

Figure 2: Process for calculating our primary metric, Machine Learning Efficacy (MLE). We train a generative model, which produces a synthetic dataset. Two downstream classifiers are trained: one on the generative model's training data and the other on the synthetic data. Each downstream model is evaluated on real test data. MLE is the difference in downstream models' test-time performance. Higher scores indicate better-quality synthetic data.

process. In Appendix E, we provide a formal definition of MLE, as well as *several more metrics* including Shape, Trend (Shi et al., 2025), Distance to Closest Record (DCR) and Membership Inference Attack (Yao et al., 2025) performance. These metrics additionally compare the synthetics' abilities to preserve trainset distributions without memorizing the trainset.

**Aggregation of results:** Our evaluation involves a comparison between 14 synthesis methods (including Tabby) across 8 datasets. So, while we do report final scores on each task individually, we would also like to understand which method *performs the best across all of the tasks in our evaluation.*

To do so, we aggregate MLE scores using *performance profile* curves (Dolan & Moré, 2002), a robust way to visually compare scores across noisy evaluations in a large number of environments. Performance profiles improve over simpler aggregation techniques, such as averaging scores or computing the average rank of methods across tasks. Specifically, performance profiles are useful when scores for different tasks might be on different scales (which can be an issue with averaging scores), and can take into account methods that are extremely close to the best-performing method on a task without dropping them a full rank (which can be problematic when averaging ranks).

To summarize these curves, we also calculate the *area under the performance profile (AUP) scores* (Roberts et al., 2022), which serve as a final ranking of methods. In short, the performance of a synthesis method across *all* eight datasets may be represented as just one performance profile curve. Methods with better performance will have higher curves and, therefore, higher AUP scores. As such, **the method with highest AUP score is considered the best overall method**. Details on performance profiles are in Appendix D.

### 3.1 Tabby versus Baseline Synthesis Methods

We begin by validating our first claim.
**Claim 1**: Plain-trained Tabby models generate higher-quality tabular data than prior approaches.

**Setup**: Table 2 lists MLE for each dataset. For classification datasets (Diabetes, Travel, Adult), the reported metric is the accuracy of the downstream random forest classifier, while for regression datasets (Abalone, Rainfall, House), we report the coefficient of determination $R^2$ of the downstream random forest regressor.

The "Non-Synthetic" row corresponds to the performance achieved by training the downstream classifier or regressor on real instead of synthetic data. We consider this row to be a performance ceiling for synthetic approaches. Any model and training technique that achieves MLE equal to or better than "Non-Synthetic" is considered to be a top-performing approach and is presented in bold.

**Results**: We find that ***Plain-trained Tabby models achieve the highest MLE in 5/8 datasets***. Further, Tabby reaches upper-bound performance on Diabetes, Travel, Adult, Magic and Shoppers, indicating that *Tabby synthetic data is a capable stand-in for real data* in similar scenarios for these datasets.

We also find that ***Plain is the best-performing technique for training tabular LLMs in almost all cases***: for all eight datasets, the highest-scoring LLM is trained using Plain. *Plain-trained Tabby MH models demonstrate the highest MLE among all LLM architectures and training styles.*

Table 2: Machine Learning Efficacy (MLE, ↑). The "Non-Synthetic" row is upper-bound performance given by real, non-synthetic data . Top results (or any higher than upper-bound) are **bolded**. The number of datasets that a model achieves top performance on are counted in the "# Best" column. An asterisk indicates that at least one of three runs did not produce valid samples. Tabby models are presented in *italic*. The best-performing Tabby model, *Plain Tabby MH DGPT2* is presented in purple and achieves best performance on 5/8 datasets. Terminology glossary in Appendix A.

| | Diabetes | Travel | Adult | Magic | Shoppers | Abalone | Rainfall | House | Best |
|---|---|---|---|---|---|---|---|---|---|
| **Non-Synthetic** | **0.73** | **0.87** | **0.85** | **0.82** | **0.88** | **0.45** | **0.54** | **0.61** | |
| CTGAN | $0.39 \pm 0.00$ | $0.43 \pm 0.33$ | $0.76 \pm 0.00$ | $0.58 \pm 0.07$ | $0.85 \pm 0.00$ | $0.01 \pm 0.01$ | $0.00 \pm 0.00$ | $0.00 \pm 0.00$ | 0 |
| CTGAN+ | $0.62 \pm 0.00$ | $0.81 \pm 0.00$ | $0.76 \pm 0.00$ | $0.64 \pm 0.00$ | $0.85 \pm 0.00$ | $0.24 \pm 0.03$ | $0.22 \pm 0.15$ | $0.55 \pm 0.00$ | 0 |
| TVAE | $0.62 \pm 0.00$ | $0.81 \pm 0.00$ | $0.81 \pm 0.01$ | $0.71 \pm 0.04$ | $0.85 \pm 0.00$ | $0.07 \pm 0.03$ | $0.00 \pm 0.00$ | $0.05 \pm 0.09$ | 0 |
| CLLM | $\mathbf{0.74 \pm 0.02}$ | $0.83 \pm 0.03$ | $0.80 \pm 0.02$ | $0.17 \pm 0.00$ | $\mathbf{0.89 \pm 0.01}$ | $0.00 \pm 0.00$ | $0.00 \pm 0.00$ | N/A* | 2 |
| TabSyn | $0.65 \pm 0.01$ | $0.74 \pm 0.13$ | $0.80 \pm 0.04$ | $0.81 \pm 0.03$ | $\mathbf{0.88 \pm 0.01}$ | $0.13 \pm 0.23$ | $0.45 \pm 0.00$ | $0.60 \pm 0.01$ | 1 |
| TabDiff | $\mathbf{0.75 \pm 0.02}$ | $0.86 \pm 0.02$ | $0.83 \pm 0.01$ | $\mathbf{0.82 \pm 0.03}$ | $\mathbf{0.89 \pm 0.01}$ | $0.41 \pm 0.01$ | $0.43 \pm 0.02$ | $\mathbf{0.61 \pm 0.00}$ | 4 |
| Forest Diffusion | $\mathbf{0.76 \pm 0.00}$ | $0.86 \pm 0.01$ | $0.81 \pm 0.00$ | $0.64 \pm 0.00$ | $0.85 \pm 0.00$ | $0.35 \pm 0.00$ | $0.45 \pm 0.02$ | $0.56 \pm 0.01$ | 1 |
| Tab-DDPM | $\mathbf{0.75 \pm 0.02}$ | $\mathbf{0.87 \pm 0.01}$ | $0.84 \pm 0.00$ | $0.81 \pm 0.00$ | $0.86 \pm 0.00$ | $0.41 \pm 0.01$ | $\mathbf{0.54 \pm 0.01}$ | $0.43 \pm 0.01$ | 3 |
| *Plain* Base | $\mathbf{0.75 \pm 0.02}$ | $0.86 \pm 0.01$ | $\mathbf{0.85 \pm 0.00}$ | $0.80 \pm 0.03$ | $\mathbf{0.89 \pm 0.01}$ | $\mathbf{0.44 \pm 0.01}$ | $0.52 \pm 0.03$ | $0.55 \pm 0.08$ | 4 |
| *Plain Tabby MH* | $\mathbf{0.74 \pm 0.00}$ | $\mathbf{0.88 \pm 0.01}$ | $\mathbf{0.85 \pm 0.00}$ | $\mathbf{0.82 \pm 0.02}$ | $\mathbf{0.89 \pm 0.01}$ | $0.43 \pm 0.01$ | $0.49 \pm 0.00$ | $0.60 \pm 0.00$ | 5 |
| GReaT Base | $0.62 \pm 0.01$ | $0.85 \pm 0.02$ | $0.83 \pm 0.01$ | $0.80 \pm 0.01$ | $0.87 \pm 0.00$ | $0.41 \pm 0.01$ | N/A* | $0.56 \pm 0.01$ | 0 |
| GReaT *Tabby MH* | $0.64 \pm 0.01$ | $0.86 \pm 0.01$ | $0.83 \pm 0.00$ | $0.81 \pm 0.01$ | $\mathbf{0.89 \pm 0.01}$ | $0.40 \pm 0.01$ | $0.00 \pm 0.00$* | $0.56 \pm 0.01$ | 1 |
| GTT Base DGPT2 | $0.72 \pm 0.06$ | $\mathbf{0.87 \pm 0.02}$ | $0.83 \pm 0.01$ | $0.79 \pm 0.01$ | $0.87 \pm 0.00$ | $0.40 \pm 0.01$ | $0.05 \pm 0.01$ | $0.55 \pm 0.02$ | 1 |
| GTT *Tabby MH* | $0.62 \pm 0.00$ | $0.85 \pm 0.01$ | $0.76 \pm 0.07$ | $0.81 \pm 0.02$ | $\mathbf{0.88 \pm 0.00}$ | $0.37 \pm 0.02$ | $0.26 \pm 0.37$ | $0.55 \pm 0.00$ | 1 |

For the Rainfall dataset, pre-existing LLM tabular training techinques introduce undesirable effects. Entries marked by an asterisk (*) for this dataset indicate that at least one of three runs were unsuccessful in synthesizing *any* valid samples. Particularly, the Non-Tabby GReaT model is unable to produce valid samples in any of the runs. Meanwhile, each Plain-trained model is successfully sampled and outperforms all GReaT or GTT-trained models in all three runs, indicating that **Plain-trained Tabby models are capable of modeling complexities within the Rainfall dataset that pre-existing LLM-based tabular synthesis works are unable to capture**.

**Performance Profile Analysis:** The performance profile curves in Figure 3 support our findings. In particular, Plain-trained Tabby MH achieves the highest AUP score. This indicates that **Plain-trained Tabby MH performs the best among all methods** when comparing across all datasets.

Further, we see that the top two synthesis approaches are the two Plain-trained models, which surpass the prior SOTA method of TabDiff. ***Given that these models rely on fewer assumptions than diffusion approaches, and are simpler to train than the GTT or GReaT LLMs, we find that both Tabby MH and Plain training are powerful advancements for the task of tabular data synthesis.***

**Comparing Multivariate Modeling Capabilities:** We further compare the multivariate modeling capabilities of Tab-DDPM, Plain-trained Tabby MH and the prior top-performing LLM-based approach of GReaT-trained Non-Tabby with TapTap and Tabula in Figure 4. We plot the House dataset's target column (Median House Value) as a function of its most predictive feature in the dataset (Median Income), for (left to right) real data, Plain Tabby MH, non-Tabby GTT and Tab-DDPM.

Tab-DDPM's plot (left) differs the most from the real data (right) because the Tab-DDPM model only supports integer-valued regression targets. Accordingly, both LLM-based approaches more accurately capture the target column's distribution than Tab-DDPM.

Meanwhile, GReaT sampling (center left) constrains that the target column distribution in the training dataset is replicated in synthetic data, by prompting the model with target values selected randomly based on their frequency in the training data. Accordingly, GReaT models will not generate target values outside those in the training data, which can be undesirable for datasets with few rows or limited target column coverage. In contrast, Plain training (center right) allows the model to generate previously unseen target

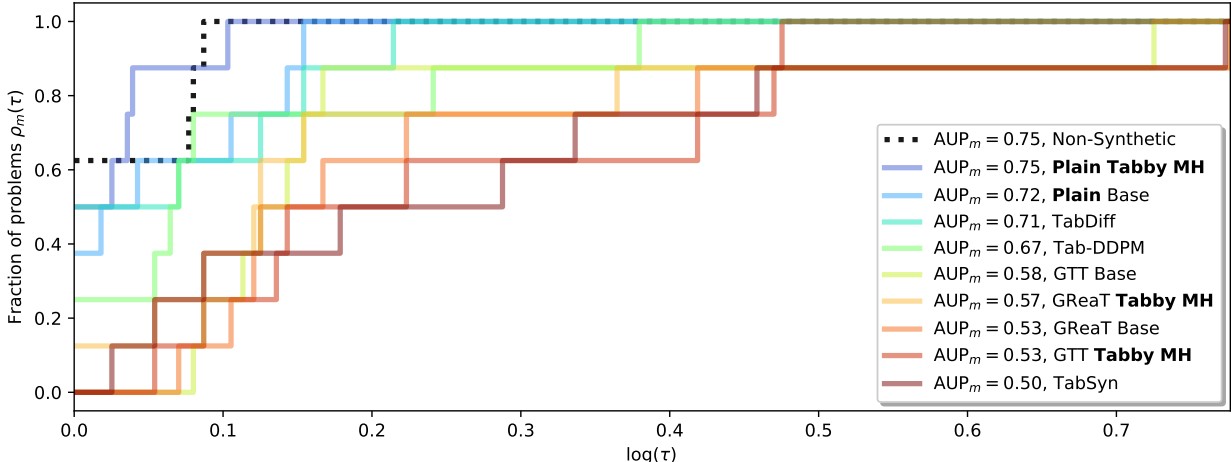

Figure 3: Performance profile curves and AUP scores across computed using the MLE scores on our evaluation tasks. The top performing method is *Tabby MH DGPT2* with Plain training.

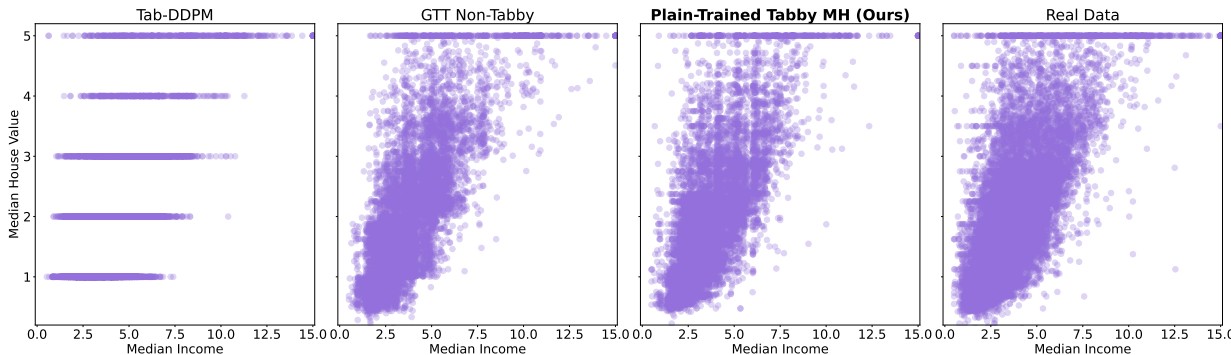

Figure 4: The House dataset's target Median House Value column as a function of its most-predictive feature, Median Income. Left to right: synthetic data from Tab-DDPM, the prior best LLM-based method and Plain Tabby MH, followed by the original data distribution. Tabby demonstrates a similar distribution to the real data and to GTT, but without GTT's assumptions (listed in Section 3.1).

values. The improved modeling capacity of Tabby over the Non-Tabby model allows Plain's sampling approach to effectively capture the overall target column distribution.

## 3.2 Investigating the Choice of Base Model

We now turn to our second claim.
**Claim 2**: The Tabby architecture modification allows smaller LLMs to achieve similar or better synthetic data fidelity than LLMs with higher parameter counts.

**Comparisons:** We compare synthesis quality across LLMs of varying sizes. We consider 7 LLMs, listed in Table 3, evaluating Non-Tabby and MH versions of each. Each model is Plain-trained under conditions provided in Section 3.0, then sampled 500 times. Results are averaged across two runs. Llama models use LoRA (Hu et al., 2021) on all linear transformer layers, with the LM head fully fine-tuned.

**Results:** Table 3 and Figure 7 display results for the Travel dataset, with results for Diabetes and House (plus additional metrics and results for GReaT training) in Appendix E.3.

We find that **Tabby improves MLE or maintains upper-bound MLE for 6/7 models**, without necessarily increasing the cost of inference (Section 2.4). Although higher-parameter models are generally correlated with greater generative abilities, Figure 7 demonstrates that this is not always the case: Interestingly, we find that the Llama models (1.2B and 8B parameters each), have lower average MLE than smaller models.

Table 3: MLE for Base and MH versions of 7 LLMs with varying parameter counts, for the Travel dataset. Results higher than Non-Synthetic are presented in **bold**. Tabby improves or maintains upper-bound MLE for 6/7 models. The parameter count of a Tabby MH model for a $V$-column dataset equals the parameter count of the base model without its head, plus $V$ times the parameter count of the base model's head. Results for Diabetes and House (and additional metrics) in Appendix E.3.

|  | MLE ($\uparrow$) | Params |
|---|---|---|
| **Non-Synthetic (Upper Bound)** | **0.87** | |
| Base Pythia 14M | $0.86 \pm 0.01$ | 14M |
| *Tabby MH Pythia 14M* | $0.82 \pm 0.02$ | 53M |
| Base Distilled-GPT2 | $\mathbf{0.88 \pm 0.00}$ | 82M |
| *Tabby MH Distilled-GPT2* | $\mathbf{0.89 \pm 0.02}$ | 310M |
| Base GPT2 | $\mathbf{0.89 \pm 0.01}$ | 120M |
| *Tabby MH GPT2* | $\mathbf{0.87 \pm 0.01}$ | 360M |
| Base Pythia 160M | $\mathbf{0.87 \pm 0.01}$ | 160M |
| *Tabby MH Pythia 160M* | $0.86 \pm 0.00$ | 390M |
| Base Pythia 410M | $0.86 \pm 0.02$ | 410M |
| *Tabby MH Pythia 410M* | $\mathbf{0.88 \pm 0.03}$ | 710M |
| Base Llama 3.2 1B | $0.82 \pm 0.01$ | 1.2B |
| *Tabby MH Llama 3.2 1B* | $0.84 \pm 0.02$ | 2.8B |
| Base Llama 3.1 8B | $0.84 \pm 0.01$ | 8.0B |
| *Tabby MH Llama 3.1 8B* | $0.86 \pm 0.03$ | 11B |

Tabby offers favorable performance improvements relative to the scaling curve and **allows even small models to better outperform large, resource-intensive models**.

### 3.3 Extending Tabby Beyond Tabular Data to General Structured Modalities

While tabular data is frequently overlooked in contemporary machine learning research, related structured modalities such as nested data receive even less attention. While GReaT, TapTap, Tabula, CTGAN and TVAE are focused solely on tabular data and do not clearly extend beyond tables, we demonstrate that Tabby can be generalized to address our third claim.

**Claim 3**: Tabby architecture modifications may also be applied to other structured data beyond tabular data, resulting in higher-quality synthetic data for these modalities as well.

**Comparisons:** We plain-train non-Tabby and Tabby MH models on a JSON dataset of patients being evaluated for Glaucoma (Manoj, 2024). Each datapoint has 10 features, organized in 3 groups: a group of 7 columns representing qualitative aspects of the optic nerve, a group of 2 columns corresponding to measurements between the optic nerve and eye, and a standalone feature for the diagnosis (examples in Box D). The binary classification target is inside the first group and assesses whether the optic nerve is thinning. As with tabular datasets in Section 3.1 and 3.2, we train downstream classifiers to predict the target variable and then present the resulting MLE.

We also consider the *discrimination* metric: Given equal numbers of real and synthetic samples, we measure the accuracy of a discrimination classifier that is trained to distinguish real versus synthetic datapoints. Because 50% accuracy would indicate that the classifier is fully unable to distinguish real from synthetic, we report the accuracy's distance from 50% in Table 4 so that *lower scores indicate higher-quality synthesis*.

**Results:** Table 4 demonstrates that Tabby MH improves MLE to parity with real data. Tabby MH's lower discrimination score signifies this model's samples are more realistic than non-Tabby samples.

Table 4: MLE and Discrimination scores for Plain-trained Base and MH models on a dataset of JSON records. Each record contains diagnostic information of a glaucoma sufferer or a healthy patient.

| | MLE ($\uparrow$) | Discrim. ($\downarrow$) |
|---|---|---|
| **Non-Synthetic (Upper Bound)** | **0.97** | |
| CTGAN | 0.52 | 0.46 |
| Forest Diffusion | 0.95 | 0.31 |
| Tab-DDPM | 0.94 | 0.45 |
| Base DGPT2 | 0.93 | 0.06 |
| *Tabby MH* DGPT2 | **0.97** | **0.01** |

### 3.4 Tracking the Adaptation to Individual Columns

We address our fourth claim by examining Tabby's progress while fine-tuning on tabular data.
**Claim 4**: Tabby's loss formulation allows for convenient tracking of per-column performance at training time, leading to better understanding of model behavior.

**Setup:** For three runs, we train a Tabby MH model on a subset of the House dataset containing 5160 rows and six columns. We log the individual columns' losses on the evaluation dataset every 2500 steps while training for 10 epochs, then average across the runs.

**Results**: Individual column losses are shown in Figure 8. This information can be vital to understanding model behavior and training progress, as elaborated in Section E.4.

### 3.5 Discussion

We find that Tabby models synthesize high-quality data in a variety of settings. In particular, **Plain-trained Tabby MH consistently outperforms all prior LLM-based approaches** and is comparable to or better than diffusion-based approaches in most settings, despite Tabby enjoying greater flexibility under fewer assumptions than those made for diffusion models. The Tabby architecture modification allows LLMs to better model both univariate column distributions and multivariate relationships across columns.

Unusually, we find that the baseline Plain training technique with Distilled-GPT2 performs quite well on several standard evaluation datasets. **The high performance of the Plain training technique compared to prior LLM works on tabular synthesis, which are more complex, is surprising.** Notably, GPT-2 and related models can even outperform much larger models such as Llama on these benchmarks. A plausible explanation is that tabular generative ability may depend on the proportion of parameters allocated to the LM head relative to the rest of the model—about 33% in GPT-2 versus roughly 7% in Llama-8B—suggesting that heavier output heads may better capture the structured dependencies characteristic of tabular data.

As of this writing, the Adult, House and Diabetes datasets have become quite prevalent for tabular synthesis evaluation. We hope that future research will build off of our evaluation setup by continuing to include more diverse and challenging tabular datasets, along with extensions to other structured modalities.

**Limitations:** We highlight two additional priorities for future work:

- Privacy preservation is important for trainsets that contain sensitive data, such as patient medical information. While Tabby's privacy preservation is similar to prior works (Appendix E), a method with strong formal privacy guarantees is an important next step for privacy-critical applications.

- Computationally-constrained environments or tasks with particularly large datasets may require deep learning approaches that are specifically designed with efficiency in mind. The Plain training method and insights from Section 3.2 may be useful towards this priority, but we leave further experimentation and the efficient implementations detailed in Section 2.4 to future work.

## 4    Related Work

Tabular data has played a central role in machine learning since the field's early days. In particular, decision trees and relatives (Song & Lu, 2015) are well-adapted to table classification or regression. Table synthesis is a growing area, though frequently overlooked in favor of image and text synthesis.

**Classical synthesis:** Classic machine learning models, such as random forest models or Bayesian networks, may be used to synthesize tables (Reiter, 2005; Zhang et al., 2017), but are limited in the data types and distributions that may be represented.

**Generative Adversarial Networks (GANs):** Many tabular synthesis methods rely on GANs (Goodfellow et al., 2014; Xu et al., 2019), but have encountered several limitations. In particular, distributions of ordinal columns are frequently imbalanced, leading GANs to undesirable phenomena such as mode collapse. Continuous columns may possess multiple modes and complex distributions, which GANs also struggle to capture (Xu et al., 2019).

**Diffusion Models**: Forest Diffusion (Jolicoeur-Martineau et al., 2024) and Tab-DDPM (Kotelnikov et al., 2022) are state-of-the-art table synthesis approaches based on the diffusion model. Both show top performance on many standard tabular metrics and are reliable for certain applications. Unfortunately, this performance is achieved by strong assumptions on the nature of the data space–for instance, numeric target variables may only assume integer values (see Figure 4) and diffusion models are unable to model non-categorical string columns such as addresses or telephone numbers. The ability to reach table synthesis performance comparable to that of diffusion models, but with fewer assumptions, is as an area of active research.

**LLMs:** A small, but growing, body of work has applied LLMs' flexible modeling abilities to tables. GReaT (Borisov et al., 2022) is a method to convert tabular data into a sentence format compatible with LLMs, then "shuffling" the order in which columns occur for each row to improve the modeling of inter-column dependencies. TapTap (Zhang et al., 2023) pretrains LMs on a variety of tabular data before fine-tuning on a downstream table synthesis task, while Tabula (Zhao et al., 2023) explores methods of preprocessing the training data to decrease sequence length. Other LLM-based works have adapted these advances to relational tables (Solatorio & Dupriez, 2023), or used the emergent abilities of very large models such as GPT-4 to generate synthetic data using In-Context Learning in place of fine-tuning (Seedat et al., 2024). Many of these works can be used in concert with Tabby, as demonstrated in Section 3, and they offer the additional advantage over other architectures of enabling pretrained tabular models to adapt effectively to new datasets.

**MoE Architectures:** The key innovation of Tabby is the application of Gated Mixture of Expert (MoE) layers (Shazeer et al., 2017; Masoudnia & Ebrahimpour, 2014) for LLM table synthesis. MoE layers have enjoyed utility in multitask (Ma et al., 2018; Gupta et al., 2022) and multimodal learning (Zhao et al., 2024; Park et al., 2018), by creating sets of model parameters dedicated to a specific task.

## 5    Conclusion

Tabby is an MOE-based architecture modification that allows LLMs to generate realistic tabular data. Tabby reaches MLE parity with real data in 5/8 datasets. We hope this promising performance spurs future work on architecture modifications that allow LLMs to represent structured data.

### Broader Impact Statement

Tabby is an improvement in the realism of synthetic tabular data, with extensions to non-tabular structured data. As such, Tabby (and its downstream applications) will have positive impacts on tasks that require synthetic structured data, such as low-data downstream tasks or privacy-critical tasks. However, Tabby may also be useful for negative downstream applications, such as the falsification of data—a drawback common to many advancements in the realism of generative modeling.

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

# A    Terminology Glossary

For convenience, in addition to definitions within the main text, we list and define the most frequently-used terms and abbreviations in our paper here:

- **DGPT2**: Distilled-GPT2 (Radford et al., 2019).

- **Distance to Closest Record (DCR)**: Metric for synthetic data quality and privacy, defined in Appendix D.

- **Discrimination**: Metric for synthetic data quality, defined in Appendix D.

- **GReaT** (Borisov et al., 2022): The landmark work on fine-tuning pre-existing LLMs to synthesize tabular data by encoding datapoints as text. Similar to Plain training, but includes train-time complications such as shuffling the order in which columns are encoded and sample-time complications such as the inability to generate label values that do not occur in the training dataset. Discussed in-detail in Section 4.

- **GReaT+Tabula (GT)**: The combination of GReaT training plus Tabula (Zhao et al., 2023) data encoding; see Section 4.

- **GReaT+TapTap+Tabula (GTT)**: The combination of GReaT training plus Tabula encoding and TapTap (Zhang et al., 2023) pre-training on tabular data (which is performed *after* the LLM is pre-trained on text data).

- **Low Rank Adapters (LoRA)**: Parameter-efficient training method from (Hu et al., 2021).

- **Mixture-of-Experts (MoE)**: Architecture technique which replaces one block with a set of specialized blocks; see Section 4.

- **Multi-Head (MH)**: The best-performing variant of Tabby, which replaces the LLM's language model output layer with an MoE layer.

- **Machine Learning Efficacy (MLE)**: Our primary evaluation metric, introduced in Section 3.0 and discussed in-detail in Appendix D.

- **Multi-MMLP (MMLP)**: Tabby modification that applies MoE to the transformer blocks' MLPs.

- **Multi-MLP and LM Head (MMLP+MH)**: Tabby modification that applies MoEs to **both** the transformer blocks' MLPs and to the language model output layers.

- **Non-Synthetic (Upper Bound)**: Used for the MLE metric, this score represents the performance of a downstream classifier trained on real, instead of synthetic, data. See Appendix D for details.

- **Non-Tabby (NT)**: An LLM without the Tabby modification, also referred to as a Base LLM.

- **Plain**: Our simple but high-performing technique for training LLMs on tabular data; introduced in Section 2.

- **Tab-DDPM (TDDPM)**: A state-of-the-art tabular synthesis technique based on the diffusion model architecture, which relies on several important assumptions; see Section 4.

# B    Additional dataset information

We select a variety of tabular datasets for our evaluations, with two goals in mind. First, the inclusion of the most standard tabular datasets—Diabetes, Adult and House—allows for easy comparison with prior works. Second, we include classification and regression datasets from a variety of domains, such as Earth science (Rainfall), business (Travel) and medicine (Diabetes). This diversity allows us to demonstrate that Tabby models achieve high performance across a variety of real-world data types and distributions. Refer to Table 5 for download links to each dataset.

*Diabetes* (Kahn, 1994) contains medical information on female hospital patients, including age, number of pregnancies and skin thickness. Downstream models learn to predict whether a given patient suffers from diabetes. Apart from the label, all dataset columns are numerical, with some columns taking only integer values, while others are floats.

Table 5: Download links for each dataset.

| Dataset | Link |
|---|---|
| Diabetes | https://www.openml.org/search?type=data&sort=runs&id=37&status=active |
| Travel | https://www.kaggle.com/datasets/tejashvi14/tour-travels-customer-churn-prediction/data |
| Adult | https://archive.ics.uci.edu/dataset/2/adult |
| Magic | https://archive.ics.uci.edu/dataset/159/magic+gamma+telescope |
| Shoppers | https://archive.ics.uci.edu/dataset/468/online+shoppers+purchasing+intention+dataset |
| Rainfall | https://www.openml.org/search?type=data&status=active&id=41539&sort=runs |
| Abalone | https://www.openml.org/search?type=data&sort=runs&id=183&status=active |
| House | https://scikit-learn.org/stable/modules/generated/sklearn.datasets.fetch_california_housing.html |
| Glaucoma | https://huggingface.co/datasets/AswanthCManoj/glaucoma_diagnosis_json_analysis |

*Travel* (Tejashvi, 2023) was collected by a travel agency wishing to predict customer churn. With the binary variable churn as the target, features include whether the traveler booked a hotel, frequent flyer status and traveler age. While most features are categorical, there are two numerical columns: traveler age and the number of times that the customer has used the travel agency in the past.

*Adult* (Becker & Kohavi, 1996) is a dataset commonly used to benchmark tabular classification algorithms. Each row contains basic information on one American adult, such as their age, years of education and marital status. For each adult, the downstream task is to predict whether their annual income is above or below $50,000. The features are a mix of categorical and numerical columns, with each numerical column taking only integer values.

The *Magic* (Bock, 2004) dataset consists of Monte Carlo–simulated observations from the Cherenkov gamma telescope, designed to differentiate high-energy gamma-ray events from background hadronic showers. Each row represents a single event characterized by 10 continuous features derived from Cherenkov image parameters, such as length, width, and asymmetry. The target variable is binary, indicating whether the event originated from a gamma signal or a hadronic background, making the dataset suitable for evaluating classification performance on physics-based data.

The *Shoppers* (Sakar & Kastro, 2018) dataset contains session-level data from an e-commerce website over a one-year period. Each record describes a single visit using 18 behavioral and contextual features, including page visit counts and durations, bounce and exit rates, visitor type, month, and weekend indicator. The target variable is binary, indicating whether the session resulted in a purchase, making the dataset suitable for evaluating classification methods on imbalanced user-behavior data.

Our first regression dataset is *Abalone* (Nash et al., 1994), which records the basic measurements of abalones, such weight and height. The target variable is the abalone's age.

The *Rainfall* (Zaman, 2018) dataset, while challenging to many LLM-based synthesis methods, contains only four columns which record historical weather data in Bangladesh. Its target variable is the amount of rainfall recorded, and the features are the year, month and weather station location.

*House* (Pace & Barry, 1997) is a standard regression dataset. Each row represents a block of houses in California during the 1990 census. The dataset records the number of households residing in the block, the block's median building age, average number of bedrooms, and other basic information. The dataset's target column is the block's median house value, which is numerical and allows us to assess Tabby's synthetic data in a regression task.

*Glaucoma* (Manoj, 2024) dataset consists of JSON records describing ophthalmic patients under consideration for a glaucoma diagnosis. Each record contains various qualitative and quantitative information about the eye, as demonstrated by the examples in Box D.

## C  Tabby for Nested Data

Figure 5 provides a visualization of the Tabby architecture used in Section E.4 to generate nested JSON data.

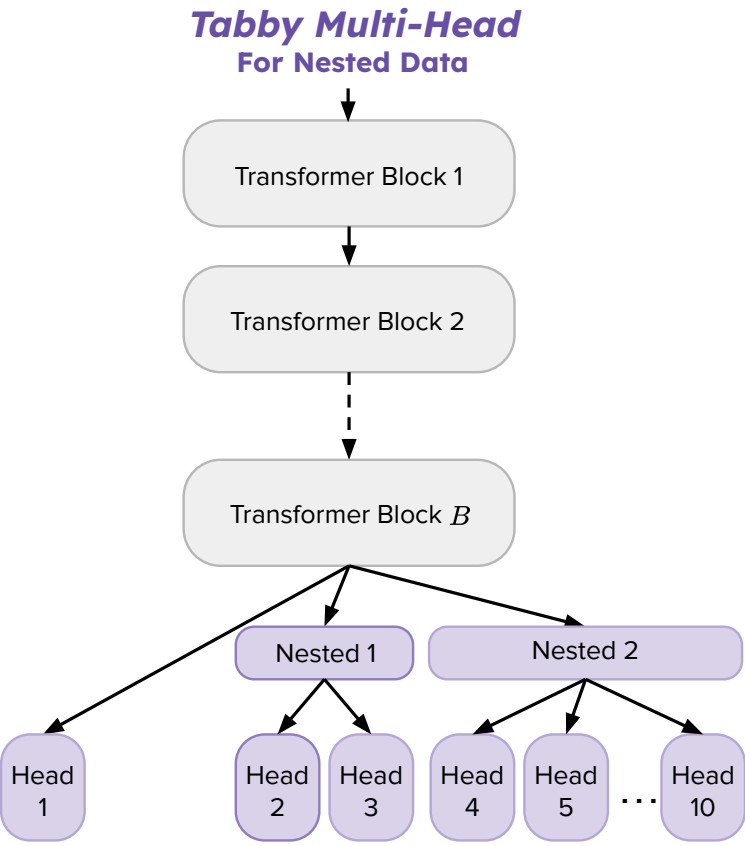

Figure 5: An overview of the Tabby MH modifications for the nested Glaucoma dataset.

## D  Details on Experimental Setup

**Calculation of results:** The reported result for each model and training setup is the average across three training runs, where not otherwise stated. For each of the three trained models, we sample $10,000$ datapoints, compute evaluation metrics separately for the three resulting synthetic datasets, then calculate the average metric value across all runs. For LLM approaches, each model is trained for up to 50 epochs, using early stopping when the validation loss (assessed every 5000 steps) fails to improve twice in a row. We perform grid search to select the learning rate with lowest validation loss for each model and training setup, with selected learning rates reported in Appendix E.6. For non-LLM works, we follow the procedures detailed in each of these works.

**More detailed definition of Machine Learning Efficacy (MLE):** Given a synthetic dataset produced by a generative model, we begin to calculate MLE by training one *downstream classifier* using the synthetic dataset. Then, we evaluate the performance of this downstream classifier on a real test set, drawn from the same distribution as the generative model's train set. We compare this classifier's performance to a *second* classifier, which is trained on the same training data as the generative model. If the synthetically-trained classifier performs worse than the classifier trained on real data, then (intuitively) the synthetic data is of lower quality than the real data: for instance, the distributions of features in the real data are not well-reflected in the synthetic data.

Put another way: given a real dataset, we form disjoint training and test sets, denoted $R$ and $D$ respectively. A generative model is trained on $R$, then generates synthetic dataset $S$.

To calculate MLE, a downstream classifier or regressor $K_R$ is trained using $R$ to predict a predetermined label column, using all other columns as features. An additional classifier or regressor $K_S$ is similarly trained on $S$. Then, the performance of $K_S$ and $K_R$ on the real test dataset $D$ is evaluated: a high-fidelity synthetic dataset

$S$ will allow $K_S$ to exhibit similar performance to $K_R$ despite never encountering real datapoints before test-time. We report both $K_R$ and $K_S$ in our results, considering MLE to be the difference in performance between $K_R$ and $K_S$.

We use a random forest classifier or regressor as our downstream model $K$. For classification datasets, we compare the accuracy of $K_R$ and $K_S$, while for regression datasets, we compare the coefficient of determination $R^2$. We define the coefficient of determination $R^2$ as $\max(1 - \frac{r}{t}, 0)$, where $r$ and $u$ are the residual sum of squares and total sum of squares, respectively. This formulation means that if a model performs worse than random guessing, its $R^2$ value will be represented as 0. For both the accuracy and $R^2$ coefficient metrics, a higher score indicates higher-quality data.

**Information on Performance Profiles:** For a given method $m \in M$, its performance profile curve is defined as

$$\rho_m(\tau) := \frac{1}{|T|} \left| \left\{ t \in T : \frac{s_{t,m}}{\min_{m' \in M}\{s_{t,m'}\}} \leq \tau \right\} \right|$$

for a set of tasks $T$ and scores $s_{t,m} : t \in T$, where lower values indicate better performance on each task. In order to satisfy the requirement that lower scores are better for the MLE metric, we set $s_{t,m} = 1 - \mathrm{MLE}_{t,m}$. Then for each method, we obtain a final score by taking the area under the curve $\rho_m(\tau)$ to obtain the AUP score as

$$\mathrm{AUP}_m = \int_0^{\tau*} \rho_m(\tau) d\log(\tau).$$

with $\tau*$ being the smallest $\tau$ such that $\rho_m(\tau) = 1$ for all methods $m \in M$, and where a higher AUP score indicates better performance.

**Discrimination:** Discrimination (Qian et al., 2023) quantifies the degree to which the generative model introduces spurious correlations or other patterns that differentiate synthetic from real data. Given the real training dataset $R$ and a synthetic dataset $S$, we sample the same number of rows from each. Next, we train a random forest classifier $C$ to discriminate between real and synthetic examples. Highest-quality synthetic data will result in 50% discrimination accuracy, indicating that $C$ is unable to distinguish between $R$ and $S$. For this reason, our reported discrimination scores are calculated as the absolute difference between 50% and the accuracy of discriminator $C$. Accordingly, lower discrimination scores represent better performance.

**Distance to Closest Record: (DCR)** Distance to Closest Record (DCR) (Lautrup et al., 2024) quantifies the distance between each synthetic datapoint and its most-similar example in the training set $R$. In addition to synthesis quality, this metric is an indication of the degree to which the model memorizes samples during training. Specifically, for each synthetic example $s \in S$, we compute its distance to every training point $r \in R$ (using L0 distance for categorical columns and L1 distance for numerical columns) and take the smallest of these distances. The overall DCR is then reported as the average of these minimum distances across all synthetic examples in SS. Lower DCR is associated with high-quality synthesis, but a DCR score of 0 implies that most synthetic examples are merely copies of training dataset points memorized during training. As such, we consider the best DCR to be the lowest nonzero score.

Box D: Representative examples from Glaucoma (Manoj, 2024)

```
[
  {
    "diagnosis": "glaucoma",
    "disc_info": {
      "disc_size": "large",
      "cup_disc_ratio": 0.8
    },
    "rim_info": {
      "rim_pallor": true,
      "rim_color": "pale",
      "bayoneting": true,
      "sharp_edge": true,
      "laminar_dot_sign": true,
      "notching": true,
      "rim_thinning": true
    }
  },
  {
    "diagnosis": "normal",
    "disc_info": {
      "disc_size": "normal",
      "cup_disc_ratio": 0.4
    },
    "rim_info": {
      "rim_pallor": false,
      "rim_color": "pink",
      "bayoneting": false,
      "sharp_edge": false,
      "laminar_dot_sign": false,
      "notching": false,
      "rim_thinning": false
    }
  },
  {
    "diagnosis": "normal",
    "disc_info": {
      "disc_size": "normal",
      "cup_disc_ratio": 0.4
    },
    "rim_info": {
      "rim_pallor": false,
      "rim_color": "pink",
      "bayoneting": false,
      "sharp_edge": false,
      "laminar_dot_sign": false,
      "notching": false,
      "rim_thinning": false
    }
  }
]
```

# E   Further Experimental Results

### E.1   Additional Metrics for Main Results Tables

For the experiment presented in Section 3.1, we include five additional metrics:

1. **Discrimination** (Table 6) works similarly to a GAN's discriminator (Goodfellow et al., 2014): the downstream discriminator model receives equal numbers of real and of synthetic datapoints, then learns to distinguish between them. The more difficulty that the discriminator model encounters in distinguishing between real and synthetic, the more we can say that the real dataset's patterns are preserved within the synthetic examples.

2. **Distance to Closest Record** (**DCR**, Table 7) measures how far the average synthetic datapoint lies from its nearest non-synthetic datapoint. If DCR equals zero, it indicates that the model has memorized its trainset, while a very large DCR indicates that the model is not preserving the trainset's patterns very well: small, non-zero, DCR scores are ideal.

3. **Shape** (Shi et al., 2025) (Table 8) measures, for each individual column, how well the synthetic column matches the real column's distribution. We follow the SDMetrics library's implementation, which uses the Kolmogorov-Smirnov statistic for categorical and the complement of Total Variation Distance for numerical columns, then averages the distances across all columns to report a final summary number.

4. **Trend** (Shi et al., 2025) (Table 9)—which is often used in conjunction with Shape—measures relationships *between* columns, by quantifying the degree to which these relationships in the real data are preserved by the synthetic data. We again use the SDMetrics library's implementation, which measures Pearson correlation for relationships between numerical columns and complement of Total Variation Distance for relationships between categorical columns or one categorical and one numerical column (which is first binned to discretize the values).

5. **Wasserstein Distance** (Table 10) computes the distance between the real and synthetic datasets' numeric columns. This metric allows us to capture the similarity of relationships among all numeric columns, as opposed to the pairwise interactions measured by the Trend metric.

6. **Membership Inference Attacks** (Yao et al., 2025) (MIA, Table 11) assess the likelihood of a synthesis method to leak the contents of its training data. We use the implementation from German (2025), which evaluates the performance of 50 different attack strategies on each synthesis method. We focus on the Abalone dataset and aggregate results by providing the maximum AUROC, mean AUROC and maximum accuracy across the 50 attacks for Tabby and other SOTA methods. For each metric, a lower number indicates a lower rate of trainset leakage and a better degree of privacy.

7. **Runtime** (Table 12) is provided for Tabby and other SOTA methods. We report the time in minutes to train for one epoch and to produce 100 samples with one NVIDIA RTX A6000. Each method is run using the out-of-the-box implementation available online.

These metrics largely corroborate our findings in Section 3.1. In particular, Plain Tabby MH's low DCR and discrimination scores indicate that this model's synthetic data closely resembles that of real data. Additionally, the DCR scores are small but *nonzero*, which indicates that the model is generating novel datapoints rather than simply repeating datapoints memorized during training.

Additionally, we provide results for the HARMONIC method (Wang et al., 2024) in Table 13.

Table 6: Discrimination metric (↓), defined in Appendix D, for approaches compared in Section 3.1. Tabby produces data with better MLE without worsening the synthetic data's discrimination score, performing competitively with Tab-DDPM.

| | Diabetes | Travel | Adult | Magic | Shoppers | Abalone | Rainfall | House |
|---|---|---|---|---|---|---|---|---|
| CTGAN | $0.42 \pm 0.00$ | $0.27 \pm 0.01$ | $0.48 \pm 0.00$ | $0.34 \pm 0.03$ | $0.50 \pm 0.00$ | $0.46 \pm 0.00$ | $0.18 \pm 0.05$ | $0.32 \pm 0.06$ |
| CTGAN+ | $0.34 \pm 0.01$ | $\mathbf{0.01 \pm 0.52}$ | $0.52 \pm 0.02$ | $0.22 \pm 0.07$ | $0.50 \pm 0.00$ | $0.02 \pm 0.47$ | $0.47 \pm 0.02$ | $0.02 \pm 0.00$ |
| TVAE | $0.45 \pm 0.02$ | $0.50 \pm 0.00$ | $0.46 \pm 0.01$ | $0.33 \pm 0.03$ | $0.50 \pm 0.00$ | $0.45 \pm 0.02$ | $0.41 \pm 0.01$ | $0.39 \pm 0.03$ |
| CLLM | $0.28 \pm 0.03$ | $0.03 \pm 0.39$ | $0.39 \pm 0.02$ | $0.30 \pm 0.00$ | $0.03 \pm 0.00$ | $0.02 \pm 0.37$ | $0.37 \pm 0.02$ | N/A* |
| TabSyn | $0.10 \pm 0.00$ | $0.00 \pm 0.44$ | $0.44 \pm 0.01$ | $0.33 \pm 0.00$ | $0.50 \pm 0.00$ | $\mathbf{0.01 \pm 0.41}$ | $0.41 \pm 0.00$ | $0.00 \pm 0.00$ |
| TabDiff | $0.33 \pm 0.19$ | $0.19 \pm 0.47$ | $0.47 \pm 0.15$ | $\mathbf{0.01 \pm 0.01}$ | $\mathbf{0.03 \pm 0.01}$ | $0.15 \pm 0.43$ | $0.43 \pm 0.12$ | $0.12 \pm 0.00$ |
| Forest Diffusion | $0.27 \pm 0.00$ | $0.28 \pm 0.00$ | $0.50 \pm 0.00$ | $\mathbf{0.01 \pm 0.01}$ | $0.04 \pm 0.01$ | $0.24 \pm 0.00$ | $0.09 \pm 0.00$ | $0.16 \pm 0.00$ |
| Tab-DDPM | $0.11 \pm 0.00$ | $0.05 \pm 0.03$ | $\mathbf{0.01 \pm 0.01}$ | $\mathbf{0.01 \pm 0.01}$ | $0.50 \pm 0.00$ | $0.03 \pm 0.01$ | $\mathbf{0.01 \pm 0.02}$ | $0.33 \pm 0.04$ |
| *Plain* Base | $\mathbf{0.04 \pm 0.01}$ | $0.03 \pm 0.02$ | $0.09 \pm 0.01$ | $0.13 \pm 0.00$ | $0.05 \pm 0.01$ | $0.06 \pm 0.01$ | $0.03 \pm 0.01$ | $0.07 \pm 0.06$ |
| *Plain Tabby MH* | $0.06 \pm 0.02$ | $0.02 \pm 0.01$ | $0.10 \pm 0.01$ | $0.12 \pm 0.01$ | $0.06 \pm 0.01$ | $0.06 \pm 0.00$ | $0.08 \pm 0.00$ | $\mathbf{0.03 \pm 0.01}$ |
| GReaT Base | $0.28 \pm 0.01$ | $0.06 \pm 0.01$ | $0.20 \pm 0.01$ | $0.22 \pm 0.00$ | $0.20 \pm 0.01$ | $0.08 \pm 0.02$ | N/A* | $0.16 \pm 0.01$ |
| GReaT *Tabby MH* | $0.29 \pm 0.02$ | $0.08 \pm 0.03$ | $0.20 \pm 0.01$ | $0.18 \pm 0.01$ | $0.17 \pm 0.00$ | $0.11 \pm 0.03$ | $0.45 \pm 0.09*$ | $0.19 \pm 0.01$ |
| GTT Base | $0.27 \pm 0.02$ | $0.07 \pm 0.01$ | $0.20 \pm 0.02$ | $0.21 \pm 0.00$ | $0.20 \pm 0.01$ | $0.05 \pm 0.01$ | $0.39 \pm 0.11$ | $0.18 \pm 0.03$ |
| GTT *Tabby MH* | $0.28 \pm 0.02$ | $0.07 \pm 0.02$ | $0.13 \pm 0.05$ | $0.18 \pm 0.01$ | $0.18 \pm 0.00$ | $0.16 \pm 0.01$ | $0.31 \pm 0.21$ | $0.20 \pm 0.01$ |

Table 7: Distance to Closest Record (DCR, $\downarrow_{>0}$), defined in Appendix D, for approaches compared in Section 3.1. Tabby MH exhibits low, nonzero scores, indicating that its synthetic examples closely resemble real data without simply copying the training data points.

| | Diabetes | Travel | Adult | Magic | Shoppers | Abalone | Rainfall | House |
|---|---|---|---|---|---|---|---|---|
| CTGAN | $0.82 \pm 0.00$ | $0.59 \pm 0.03$ | $1.70 \pm 0.09$ | $0.75 \pm 0.07$ | $1.52 \pm 0.10$ | $0.76 \pm 0.02$ | $0.03 \pm 0.01$ | $0.13 \pm 0.02$ |
| CTGAN+ | $0.52 \pm 0.03$ | $0.52 \pm 0.04$ | $3.24 \pm 0.66$ | $0.35 \pm 0.13$ | $1.8 \pm 0.30$ | $0.26 \pm 0$ | $0.03 \pm 0.01$ | $0.07 \pm 0.00$ |
| TVAE | $0.27 \pm 0.01$ | $0.10 \pm 0.06$ | $0.16 \pm 0.03$ | $0.21 \pm 0.00$ | $0.05 \pm 0.00$ | $0.41 \pm 0.01$ | $0.03 \pm 0.00$ | $0.07 \pm 0.00$ |
| CLLM | $0.4 \pm 0.01$ | $0.39 \pm 0.05$ | $1.84 \pm 0.68$ | $0.32 \pm 0.01$ | $0.00 \pm 0.00$ | $0.14 \pm 0.01$ | $0.04 \pm 0.02$ | N/A* |
| TabSyn | $0.47 \pm 0.14$ | $0.4 \pm 0.3$ | $1.29 \pm 0.77$ | $0.28 \pm 0.00$ | $1.79 \pm 0.03$ | $0.38 \pm 0.26$ | $0.01 \pm 0.00$ | $0.06 \pm 0.00$ |
| TabDiff | $0.43 \pm 0.03$ | $0.07 \pm 0.01$ | $0.38 \pm 0.03$ | $0.25 \pm 0.01$ | $0.62 \pm 0.01$ | $0.11 \pm 0$ | $0.01 \pm 0.00$ | $0.06 \pm 0.00$ |
| Forest Diffusion | $0.29 \pm 0.00$ | $0.06 \pm 0.00$ | $0.35 \pm 0.02$ | $0.24 \pm 0.00$ | $0.66 \pm 0.03$ | $0.09 \pm 0.01$ | $\mathbf{0.01 \pm 0.00}$ | $0.06 \pm 0.00$ |
| Tab-DDPM | $0.63 \pm 0.04$ | $0.00 \pm 0.00$ | $0.31 \pm 0.03$ | $0.19 \pm 0.01$ | $3.9 \pm 0.34$ | $0.12 \pm 0.01$ | $\mathbf{0.01 \pm 0.00}$ | $0.08 \pm 0.00$ |
| *Plain* Base | $\mathbf{0.01 \pm 0.00}$ | $\mathbf{0.01 \pm 0.00}$ | $0.33 \pm 0.15$ | $\mathbf{0.08 \pm 0.00}$ | $\mathbf{0.04 \pm 0.01}$ | $\mathbf{0.01 \pm 0.00}$ | $0.00 \pm 0.00$ | $\mathbf{0.03 \pm 0.01}$ |
| *Plain Tabby MH* | $0.02 \pm 0.00$ | $\mathbf{0.01 \pm 0.00}$ | $0.25 \pm 0.03$ | $0.10 \pm 0.03$ | $0.08 \pm 0.03$ | $\mathbf{0.01 \pm 0.01}$ | $\mathbf{0.01 \pm 0.00}$ | $0.04 \pm 0.00$ |
| GReaT Base | $0.33 \pm 0.00$ | $0.02 \pm 0.01$ | $\mathbf{0.12 \pm 0.03}$ | $0.18 \pm 0.00$ | $0.40 \pm 0.17$ | $0.10 \pm 0.00$ | N/A* | $0.06 \pm 0.00$ |
| GReaT *Tabby MH* | $0.36 \pm 0.00$ | $\mathbf{0.01 \pm 0.00}$ | $0.17 \pm 0.08$ | $0.19 \pm 0.00$ | $0.36 \pm 0.01$ | $0.10 \pm 0.01$ | $\mathbf{0.01}*$ | $0.06 \pm 0.00$ |
| GTT Base | $0.31 \pm 0.01$ | $0.02 \pm 0.00$ | $0.14 \pm 0.01$ | $0.18 \pm 0.01$ | $0.32 \pm 0.01$ | $0.10 \pm 0.00$ | $0.02 \pm 0.00$ | $0.06 \pm 0.00$ |
| GTT *Tabby MH* | $0.37 \pm 0.01$ | $0.02 \pm 0.00$ | $0.16 \pm 0.07$ | $0.19 \pm 0.00$ | $0.39 \pm 0.03$ | $0.10 \pm 0.00$ | $0.00 \pm 0.01$ | $0.05 \pm 0.00$ |

Table 8: Shape (Shi et al., 2025) (↑), for diffusion and LLM approaches compared in Section 3.1. Tabby Plain is a best-performing method on 3 datasets–similar to the performance of diffusion-based techniques, but without the same limiting assumptions on the nature of the dataset.

| | Diabetes | Travel | Adult | Magic | Shoppers | Abalone | Rainfall | House |
|---|---|---|---|---|---|---|---|---|
| TabSyn | $0.79 \pm 0.14$ | $0.82 \pm 0.13$ | $0.81 \pm 0.15$ | $\mathbf{0.99 \pm 0.00}$ | $\mathbf{0.98 \pm 0.00}$ | $0.84 \pm 0.12$ | $\mathbf{0.98 \pm 0.00}$ | $\mathbf{0.99 \pm 0.00}$ |
| TabDiff | $0.96 \pm 0.00$ | $0.97 \pm 0.00$ | $\mathbf{0.99 \pm 0.00}$ | $\mathbf{0.99 \pm 0.00}$ | $\mathbf{0.98 \pm 0.00}$ | $0.98 \pm 0.00$ | $\mathbf{0.98 \pm 0.00}$ | $\mathbf{0.99 \pm 0.00}$ |
| Forest Diffusion | $0.91 \pm 0.00$ | $0.95 \pm 0.00$ | $0.89 \pm 0.00$ | $0.92 \pm 0.00$ | $0.68 \pm 0.00$ | $0.96 \pm 0.02$ | $0.95 \pm 0.00$ | $0.94 \pm 0.00$ |
| Tab-DDPM | $0.89 \pm 0.00$ | $\mathbf{0.99 \pm 0.00}$ | $\mathbf{0.99 \pm 0.00}$ | $\mathbf{0.99 \pm 0.00}$ | $0.63 \pm 0.03$ | $\mathbf{0.99 \pm 0.00}$ | $\mathbf{0.98 \pm 0.00}$ | $0.95 \pm 0.00$ |
| *Plain* Base | $0.96 \pm 0.02$ | $\mathbf{0.99 \pm 0.00}$ | $0.95 \pm 0.00$ | $0.85 \pm 0.00$ | $0.97 \pm 0.00$ | $0.94 \pm 0.00$ | $0.96 \pm 0.01$ | $0.95 \pm 0.03$ |
| *Plain Tabby MH* | $\mathbf{0.98 \pm 0.00}$ | $\mathbf{0.99 \pm 0.00}$ | $0.95 \pm 0.01$ | $0.85 \pm 0.00$ | $\mathbf{0.98 \pm 0.00}$ | $0.93 \pm 0.02$ | $0.93 \pm 0.00$ | $0.97 \pm 0.00$ |
| GReaT Base | $0.81 \pm 0.01$ | $0.94 \pm 0.00$ | $0.89 \pm 0.01$ | $0.85 \pm 0.00$ | $0.85 \pm 0.00$ | $0.95 \pm 0.01$ | *N/A*∗ | $0.92 \pm 0.00$ |
| GReaT *Tabby MH* | $0.85 \pm 0.00$ | $0.93 \pm 0.01$ | $0.90 \pm 0.01$ | $0.87 \pm 0.01$ | $0.88 \pm 0.00$ | $0.91 \pm 0.03$ | $0.23 \pm 0.40*$ | $0.89 \pm 0.00$ |
| GTT Base | $0.80 \pm 0.00$ | $0.93 \pm 0.00$ | $0.89 \pm 0.02$ | $0.85 \pm 0.00$ | $0.85 \pm 0.00$ | $0.95 \pm 0.00$ | $0.50 \pm 0.43*$ | $0.91 \pm 0.02$ |
| GTT *Tabby MH* | $0.83 \pm 0.01$ | $0.94 \pm 0.00$ | $0.81 \pm 0.09$ | $0.86 \pm 0.00$ | $0.88 \pm 0.00$ | $0.89 \pm 0.00$ | $0.47 \pm 0.45$ | $0.89 \pm 0.00$ |

Table 9: Trend (Shi et al., 2025) (↑), for diffusion and LLM approaches compared in Section 3.1. Tabby Plain is a best-performing method on 3 datasets, which is similar to other SOTA approaches.

| | Diabetes | Travel | Adult | Magic | Shopping | Abalone | Rainfall | House |
|---|---|---|---|---|---|---|---|---|
| TabSyn | $0.84 \pm 0.12$ | $0.68 \pm 0.22$ | $0.61 \pm 0.33$ | $\mathbf{0.99 \pm 0.00}$ | $\mathbf{0.98 \pm 0.00}$ | $0.69 \pm 0.25$ | $\mathbf{0.95 \pm 0.00}$ | $\mathbf{0.99 \pm 0.00}$ |
| TabDiff | $\mathbf{0.97 \pm 0.00}$ | $0.92 \pm 0.01$ | $\mathbf{0.97 \pm 0.01}$ | $\mathbf{0.99 \pm 0.00}$ | $\mathbf{0.98 \pm 0.00}$ | $\mathbf{0.97 \pm 0.00}$ | $0.94 \pm 0.00$ | $\mathbf{0.99 \pm 0.00}$ |
| Forest Diffusion | $0.87 \pm 0.00$ | $0.68 \pm 0.00$ | $0.60 \pm 0.00$ | $0.85 \pm 0.00$ | $0.38 \pm 0.00$ | $0.88 \pm 0.01$ | $0.58 \pm 0.00$ | $\mathbf{0.99 \pm 0.00}$ |
| Tab-DDPM | $0.88 \pm 0.00$ | $0.97 \pm 0.00$ | $\mathbf{0.97 \pm 0.00}$ | $0.97 \pm 0.01$ | $0.58 \pm 0.03$ | $\mathbf{0.95 \pm 0.00}$ | $\mathbf{0.95 \pm 0.00}$ | $\mathbf{0.99 \pm 0.00}$ |
| *Plain* Base | $0.95 \pm 0.04$ | $0.73 \pm 0.06$ | $0.88 \pm 0.04$ | $0.68 \pm 0.15$ | $0.96 \pm 0.01$ | $0.91 \pm 0.05$ | $0.89 \pm 0.01$ | $0.96 \pm 0.02$ |
| *Plain Tabby MH* | $\mathbf{0.97 \pm 0.00}$ | $\mathbf{0.98 \pm 0.00}$ | $0.88 \pm 0.02$ | $0.86 \pm 0.04$ | $0.96 \pm 0.00$ | $0.94 \pm 0.01$ | $0.87 \pm 0.00$ | $\mathbf{0.99 \pm 0.00}$ |
| GReaT Base | $0.86 \pm 0.01$ | $0.84 \pm 0.10$ | $0.77 \pm 0.02$ | $0.91 \pm 0.01$ | $0.87 \pm 0.01$ | $0.92 \pm 0.00$ | N/A* | $0.95 \pm 0.01$ |
| GReaT *Tabby MH* | $0.85 \pm 0.02$ | $0.89 \pm 0.01$ | $0.77 \pm 0.06$ | $0.90 \pm 0.03$ | $0.90 \pm 0.00$ | $0.92 \pm 0.01$ | $0.47 \pm 0.06*$ | $0.95 \pm 0.01$ |
| GTT Base | $0.88 \pm 0.01$ | $0.84 \pm 0.10$ | $0.79 \pm 0.03$ | $0.91 \pm 0.01$ | $0.88 \pm 0.01$ | $0.93 \pm 0.00$ | $0.52 \pm 0.17*$ | $0.95 \pm 0.02$ |
| GTT *Tabby MH* | $0.86 \pm 0.01$ | $0.85 \pm 0.10$ | $0.56 \pm 0.22$ | $0.88 \pm 0.01$ | $0.91 \pm 0.00$ | $0.91 \pm 0.01$ | $0.48 \pm 0.33$ | $0.96 \pm 0.01$ |

Table 10: Wasserstein distance (↓), for diffusion and LLM approaches compared in Section 3.1. Plain Tabby is the best-performing method on Diabetes, House and Shopping.

| | Diabetes | Travel | Adult | Magic | Shopping | Abalone | Rainfall | House |
|---|---|---|---|---|---|---|---|---|
| TabSyn | $75.27 \pm 47.46$ | $1.44 \pm 1.07$ | $4.1E4 \pm 3.2E4$ | $37.21 \pm 0.55$ | $278.13 \pm 65.57$ | $1.69 \pm 1.84$ | $25.05 \pm 8.58$ | $132.33 \pm 51.19$ |
| TabDiff | $26.83 \pm 2.01$ | $0.36 \pm 0.04$ | $1.1E4 \pm 4.6E3$ | $38.80 \pm 1.93$ | $240.57 \pm 74.74$ | $0.41 \pm 0.02$ | $19.71 \pm 13.29$ | $115.99 \pm 45.71$ |
| Forest Diffusion | $19.08 \pm 0.48$ | $0.35 \pm 0.07$ | $9.9E3 \pm 3.6E3$ | $38.20 \pm 0.72$ | $274.67 \pm 64.94$ | $0.43 \pm 0.10$ | $28.69 \pm 5.6$ | $110.40 \pm 20.50$ |
| Tab-DDPM | $80.54 \pm 8.09$ | $0.30 \pm 0.03$ | $\mathbf{8.3E3 \pm 1.9E3}$ | $36.45 \pm 1.00$ | $3.1E4 \pm 1.7E3$ | $0.37 \pm 0.02$ | $\mathbf{18.47 \pm 6.88}$ | $104.47 \pm 31.47$ |
| *Plain* Base | $20.83 \pm 11.04$ | $\mathbf{0.23 \pm 0.05}$ | $1.3E4 \pm 7.6E3$ | $52.71 \pm 2.33$ | $252.94 \pm 115.00$ | $0.61 \pm 0.17$ | $22.56 \pm 6.06$ | $116.17 \pm 68.63$ |
| *Plain Tabby MH* | $\mathbf{14.75 \pm 1.32}$ | $0.29 \pm 0.05$ | $1.2E4 \pm 2.9E3$ | $56.64 \pm 1.63$ | $\mathbf{180.40 \pm 87.97}$ | $0.52 \pm 0.22$ | $32.95 \pm 5.77$ | $\mathbf{91.48 \pm 9.08}$ |
| GReaT Base | $82.75 \pm 3.86$ | $0.50 \pm 0.12$ | $2.6E4 \pm 8.5E3$ | $57.88 \pm 2.78$ | $688.17 \pm 59.05$ | $0.33 \pm 0.08$ | N/A* | $387.83 \pm 28.45$ |
| GReaT *Tabby MH* | $82.69 \pm 2.09$ | $0.46 \pm 0.04$ | $3.2E4 \pm 1.7E4$ | $55.80 \pm 4.16$ | $644.95 \pm 86.52$ | $0.34 \pm 0.07$ | $7E6 \pm 6E6*$ | $304.11 \pm 63.08$ |
| GTT Base | $81.34 \pm 3.90$ | $0.48 \pm 0.03$ | $2.2E4 \pm 5.9E3$ | $54.87 \pm 0.61$ | $733.93 \pm 95.75$ | $\mathbf{0.30 \pm 0.06}$ | $3E6 \pm 6E6*$ | $309.51 \pm 78.00$ |
| GTT *Tabby MH* | $78.70 \pm 1.73$ | $0.37 \pm 0.01$ | $4.9E4 \pm 1.8E4$ | $52.89 \pm 6.76$ | $562.43 \pm 69.22$ | $0.45 \pm 0.07$ | $3E6 \pm 6E6$ | $375.05 \pm 20.07$ |

Table 11: Membership Inference Attack (MIA) performance, for SOTA approaches compared in Section 3.1. Scores are aggregated across 50 attack techniques by reporting maximum AUROC (↓), mean AUROC (↓) and maximum accuracy (↓). We find that Tabby performs similarly to most other methods.

| | TabDiff | CTGAN+ | Plain Base | Plain Tabby | GTT Base | CLLM | HARMONIC |
|---|---|---|---|---|---|---|---|
| Max AUROC (↓) | 53.40 | 100.00 | 55.50 | 55.40 | 54.00 | 56.00 | 58.10 |
| Mean AUROC (↓) | 51.03 | 52.66 | 52.43 | 52.45 | 51.64 | 53.42 | 51.45 |
| Max Acc. (↓) | 52.60 | 100.00 | 54.20 | 54.20 | 52.90 | 54.80 | 62.50 |

Table 12: Runtime on the Abalone dataset, for SOTA approaches compared in Section 3.1. "Training" reports the minutes to train one epoch, while "Synthesis" reports the minutes to produce 10 samples.

| | Plain Tabby | GReaT Base | GTT Base | HARMONIC | CTGAN | CGAN+ | CLLM | TabDiff | TabSyn |
|---|---|---|---|---|---|---|---|---|---|
| Training | 2:05 | 2:01 | 2:03 | 24:25 | 0:01 | 0:01 | N/A | 0:02 | 0:02 |
| Synthesis | 0:10 | 0:10 | 0:11 | 7:30 | 0:01 | 0:01 | 0:04 | 0:00 | 0:00 |

Table 13: Performance of the HARMONIC synthesis method (Wang et al., 2024) by various metrics.

| | Diabetes | Travel | Adult | Abalone | Rainfall | House |
|---|---|---|---|---|---|---|
| MLE | 0.64 | 0.83 | 0.76 | 0.32 | 0.54 | 0.61 |
| Discrimination | 0.00 | 0.00 | 0.00 | 0.00 | 0.00 | 0.00 |
| DCR | 0.39 | 0.03 | 0.59 | 0.12 | 0.03 | 0.08 |
| Shape | 0.71 | 0.95 | 0.80 | 0.85 | 0.53 | 0.68 |
| Trend | 0.85 | 0.74 | 0.63 | 0.91 | 0.44 | 0.97 |
| Wasserstein | 49.81 | 0.99 | 61194.57 | 0.84 | 113.87 | 559.72 |

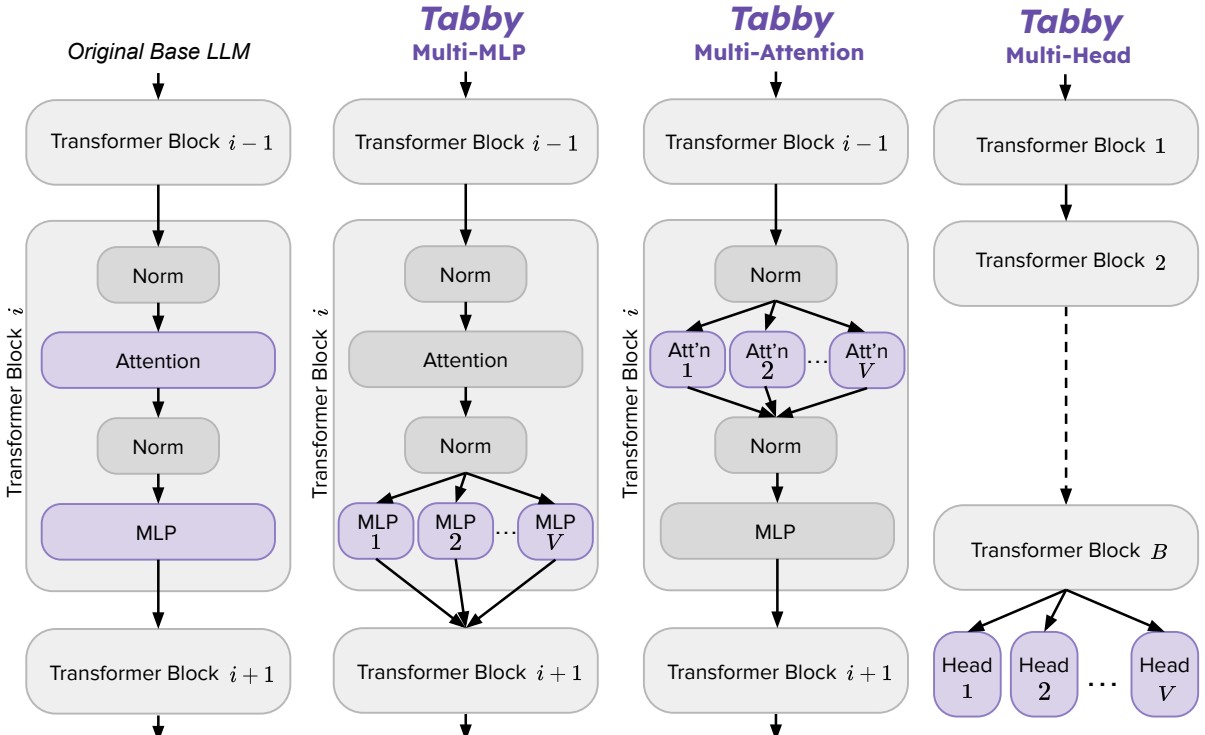

Figure 6: An overview of the Tabby MH modifications that can occur inside the LLM transformer blocks. Left to right: an original, non-Tabby LLM, a Tabby LLM with MoE MLP block, a Tabby LLM with MoE attention block, and a Tabby LLM with both MoE MLP and attention blocks. Tabby is very flexible, so as to accommodate a wide variety of tabular datasets.

### E.2 Applying Tabby to Transformer MLPs or Attention Blocks

We examine in-detail the performance of Tabby models with MoE applied to the transformer MLPs or attention blocks. We use the following terminology to refer to these architectures, visualized in Figure 6:

- **Multi-MLP** when each transformer's MLP block is replaced with an MoE layer,

- **Multi-MLP and Multi-Head (MMLP+MH)** when each transformer's MLP block is replaced with an MoE layer *and* the LM head is replaced with an MoE layer,

- **Multi-Attention (MA)** when each transformer's attention block is replaced with an MoE layer.

We focus on Tabby MH in Sections 3.1-3.4 because it demonstrates top performance in most settings. We display results for the MMLP and MMLP+MH architectures across six datasets for MLE, discrimination and DCR in Tables 14, 15 and 16, respectively. All three metrics are displayed for the MA architecture on two datasets in Table 17.

### E.3 Additional Metrics for Experiment Applying Tabby MH to Models of Varying Sizes

The results in Section 3.2 compare the MLE scores of Plain-trained models of varying sizes on the Travel dataset. Table 18 incorporates the results for the Diabetes and House datasets as well. Similarly, Table 19 presents results for models trained using GReaT and Tabula (TapTap is not included here, because TapTap-pretrained checkpoints are available only for Distill-GPT2 and GPT2).

Table 14: Machine Learning Efficacy (MLE, ↑), defined in Section 3.0, for Base non-Tabby GPT2 models, as well as Tabby models with MoE layers applied to the transformer MLPs, language modeling head, or both (notated as MMLP, MH, and MMLP+MH respectively) on select datasets.

| | Diabetes | Travel | Adult | Abalone | Rainfall | House |
|---|---|---|---|---|---|---|
| **Non-Synthetic** | **0.73** | **0.87** | **0.85** | **0.45** | **0.54** | **0.61** |
| *Plain* Non-Tabby | **0.75 ± 0.02** | 0.86 ± 0.01 | **0.85 ± 0.00** | **0.44 ± 0.01** | 0.52 ± 0.03 | 0.55 ± 0.08 |
| *Plain Tabby MMLP* | **0.75 ± 0.03** | 0.83 ± 0.02 | 0.77 ± 0.01 | 0.32 ± 0.03 | 0.35 ± 0.04 | 0.00 ± 0.00 |
| *Plain Tabby MH* | **0.74 ± 0.00** | **0.88 ± 0.01** | **0.85 ± 0.00** | 0.43 ± 0.01 | 0.49 ± 0.00 | **0.60 ± 0.00** |
| *Plain Tabby MMLP+MH* | 0.68 ± 0.02 | 0.83 ± 0.01 | 0.76 ± 0.01 | 0.33 ± 0.03 | 0.36 ± 0.19 | 0.02 ± 0.03 |
| GReaT Non-Tabby | 0.62 ± 0.01 | 0.85 ± 0.02 | 0.83 ± 0.01 | 0.41 ± 0.01 | N/A* | 0.56 ± 0.01 |
| GReaT *Tabby MMLP* | **0.74 ± 0.01** | 0.85 ± 0.03 | 0.84 ± 0.01 | 0.38 ± 0.01 | 0.24 ± 0.25 | 0.56 ± 0.02 |
| GReaT *Tabby MH* | 0.64 ± 0.01 | 0.86 ± 0.01 | 0.83 ± 0.00 | 0.40 ± 0.01 | 0.00 ± 0.00* | 0.56 ± 0.01 |
| GReaT *Tabby MMLP+MH* | 0.69 ± 0.04 | 0.83 ± 0.02 | 0.83 ± 0.01 | 0.38 ± 0.03 | 0.17 ± 0.30 | 0.57 ± 0.01 |
| GTT Non-Tabby | 0.72 ± 0.06 | **0.87 ± 0.02** | 0.83 ± 0.01 | 0.40 ± 0.01 | 0.05 ± 0.01 | 0.55 ± 0.02 |
| GTT *Tabby MMLP* | 0.69 ± 0.04 | **0.87 ± 0.01** | 0.84 ± 0.00 | 0.36 ± 0.01 | 0.03 ± 0.00* | 0.56 ± 0.01 |
| GTT *Tabby MH* | 0.62 ± 0.00 | 0.85 ± 0.01 | 0.76 ± 0.07 | 0.37 ± 0.02 | 0.26 ± 0.37 | 0.55 ± 0.00 |
| GTT *Tabby MMLP+MH* | 0.70 ± 0.04 | 0.85 ± 0.02 | 0.84 ± 0.00 | 0.38 ± 0.02 | 0.09 ± 0.13 | 0.57 ± 0.00 |

Table 15: Discrimination metric (↓), defined in Appendix D, for Base non-Tabby GPT2 models, as well as Tabby models with MoE layers applied to the transformer MLPs, language modeling head, or both (notated as MMLP, MH, and MMLP+MH respectively) on select datasets.

| | Diabetes | Travel | Adult | Abalone | Rainfall | House |
|---|---|---|---|---|---|---|
| *Plain* Non-Tabby | **0.04 ± 0.01** | 0.03 ± 0.02 | 0.09 ± 0.01 | 0.06 ± 0.01 | 0.03 ± 0.01 | 0.07 ± 0.06 |
| *Plain Tabby MMLP* | 0.22 ± 0.03 | **0.02 ± 0.02** | 0.22 ± 0.06 | 0.19 ± 0.04 | 0.12 ± 0.00 | 0.19 ± 0.06 |
| *Plain Tabby MH* | 0.06 ± 0.02 | **0.02 ± 0.01** | 0.10 ± 0.01 | 0.06 ± 0.00 | 0.08 ± 0.00 | **0.03 ± 0.01** |
| *Plain Tabby MMLP+MH* | 0.19 ± 0.02 | 0.03 ± 0.02 | 0.25 ± 0.11 | 0.22 ± 0.03 | 0.12 ± 0.01 | 0.23 ± 0.03 |
| GReaT Non-Tabby | 0.28 ± 0.01 | 0.06 ± 0.01 | 0.20 ± 0.01 | 0.08 ± 0.02 | N/A* | 0.16 ± 0.01 |
| GReaT *Tabby MMLP* | 0.23 ± 0.01 | 0.10 ± 0.02 | 0.19 ± 0.00 | 0.08 ± 0.01 | 0.27 ± 0.17 | 0.16 ± 0.01 |
| GReaT *Tabby MH* | 0.29 ± 0.02 | 0.08 ± 0.03 | 0.20 ± 0.01 | 0.11 ± 0.03 | 0.45 ± 0.09* | 0.19 ± 0.01 |
| GReaT *Tabby MMLP+MH* | 0.24 ± 0.01 | 0.09 ± 0.01 | 0.21 ± 0.01 | 0.07 ± 0.00 | 0.24 ± 0.17 | 0.16 ± 0.00 |
| GTT Non-Tabby | 0.27 ± 0.02 | 0.07 ± 0.01 | 0.20 ± 0.02 | 0.05 ± 0.01 | 0.39 ± 0.11 | 0.18 ± 0.03 |
| GTT *Tabby MMLP* | 0.28 ± 0.01 | 0.09 ± 0.01 | 0.18 ± 0.01 | 0.14 ± 0.02 | 0.46 ± 0.07* | 0.18 ± 0.01 |
| GTT *Tabby MH* | 0.28 ± 0.02 | 0.07 ± 0.02 | 0.13 ± 0.05 | 0.16 ± 0.01 | 0.31 ± 0.21 | 0.20 ± 0.01 |
| GTT *Tabby MMLP+MH* | 0.24 ± 0.01 | 0.08 ± 0.01 | 0.18 ± 0.00 | 0.14 ± 0.02 | 0.24 ± 0.24 | 0.16 ± 0.01 |

Table 16: Distance to Closest Record (DCR, $\downarrow_{>0}$), defined in Appendix D, for Base non-Tabby GPT2 models, as well as Tabby models with MoE layers applied to the transformer MLPs, language modeling head, or both (notated as MMLP, MH, and MMLP+MH respectively) on select datasets.

| | Diabetes | Travel | Adult | Abalone | Rainfall | House |
|---|---|---|---|---|---|---|
| *Plain* Non-Tabby | **0.01 ± 0.00** | **0.01 ± 0.00** | 0.33 ± 0.15 | **0.01 ± 0.00** | 0.00 ± 0.00 | **0.03 ± 0.01** |
| *Plain Tabby MMLP* | 0.35 ± 0.03 | 0.08 ± 0.00 | 0.55 ± 0.09 | 0.21 ± 0.02 | 0.03 ± 0.00 | $1.7e12 ± 2.7e12$ |
| *Plain Tabby MH* | 0.02 ± 0.00 | **0.01 ± 0.00** | 0.25 ± 0.03 | **0.01 ± 0.01** | 0.01 ± 0.00 | 0.04 ± 0.00 |
| *Plain Tabby MMLP+MH* | 0.34 ± 0.02 | 0.07 ± 0.00 | 0.39 ± 0.15 | 0.20 ± 0.03 | 0.03 ± 0.01 | $2.3e12 ± 4.1e12$ |
| GReaT Non-Tabby | 0.33 ± 0.00 | 0.02 ± 0.01 | 0.12 ± 0.03 | 0.10 ± 0.00 | N/A* | 0.06 ± 0.00 |
| GReaT *Tabby MMLP* | 0.34 ± 0.01 | 0.02 ± 0.00 | 0.12 ± 0.01 | 0.10 ± 0.00 | 0.00 ± 0.01 | 0.06 ± 0.00 |
| GReaT *Tabby MH* | 0.36 ± 0.00 | **0.01 ± 0.00** | 0.17 ± 0.08 | 0.10 ± 0.01 | **0.01*** | 0.06 ± 0.00 |
| GReaT *Tabby MMLP+MH* | 0.33 ± 0.02 | 0.02 ± 0.00 | **0.11 ± 0.01** | 0.10 ± 0.00 | **0.01 ± 0.00** | 0.06 ± 0.00 |
| GTT Non-Tabby | 0.31 ± 0.01 | 0.02 ± 0.00 | 0.14 ± 0.01 | 0.10 ± 0.00 | 0.02 ± 0.00 | 0.06 ± 0.00 |
| GTT *Tabby MMLP* | 0.31 ± 0.02 | 0.02 ± 0.00 | 0.14 ± 0.03 | 0.10 ± 0.00 | **0.01*** | 0.06 ± 0.00 |
| GTT *Tabby MH* | 0.37 ± 0.01 | 0.02 ± 0.00 | 0.16 ± 0.07 | 0.10 ± 0.00 | 0.00 ± 0.01 | 0.05 ± 0.00 |
| GTT *Tabby MMLP+MH* | 0.31 ± 0.00 | 0.02 ± 0.00 | **0.11 ± 0.02** | 0.11 ± 0.00 | **0.01 ± 0.01** | 0.06 ± 0.00 |

Table 17: All evaluation metrics, for non-Tabby models and Tabby models with MoE applied to the transformer attention blocks (abbreviated as Tabby MA) on select datasets. Base LLM is DGPT2.

|  | MLE (↑) | | Discrimination (↓) | | DCR (↓$_{>0}$) | |
|  | Diabetes | House | Diabetes | House | Diabetes | House |
|---|---|---|---|---|---|---|
| **Non-Synthetic (Upper Bound)** | **0.73** | **0.61** | | | | |
| *Plain* Non-Tabby | **0.75** | 0.55 | **0.04** | 0.07 | **0.01** | 0.03 |
| *Plain Tabby MA* | 0.62 | 0.08 | 0.23 | 0.28 | 0.41 | 0.08 |
| GTT Non-Tabby | 0.72 | 0.55 | 0.27 | 0.18 | 0.31 | 0.06 |
| GTT *Tabby MA* | 0.62 | 0.56 | 0.31 | 0.17 | 0.36 | 0.06 |

Table 18: Results using Plain training for all three datasets of the experiment in Section 3.2, which compares non-Tabby and Tabby MH models across base LLMs of varying sizes.

|  | Travel | | Diabetes | | House | |
|  | MLE (↑) | Params | MLE (↑) | Params | MLE (↑) | Params |
|---|---|---|---|---|---|---|
| **Non-Synthetic (Upper Bound)** | **0.87** | | **0.73** | | **0.61** | |
| Base Pythia 14m | $0.86 \pm 0.01$ | 14M | $\mathbf{0.76 \pm 0.02}$ | 14M | $0.52 \pm 0.07$ | 14M |
| Tabby MH Pythia 14m | $0.82 \pm 0.02$ | 53M | $\mathbf{0.77 \pm 0.00}$ | 66M | $0.54 \pm 0.01$ | 66M |
| Base Distilled-GPT2 | $\mathbf{0.88 \pm 0.00}$ | 82M | $0.73 \pm 0.02$ | 82M | $0.53 \pm 0.10$ | 82M |
| Tabby MH Distilled-GPT2 | $\mathbf{0.89 \pm 0.02}$ | 310M | $0.73 \pm 0.01$ | 390M | $\mathbf{0.61 \pm 0.01}$ | 390M |
| Base GPT2 | $\mathbf{0.89 \pm 0.01}$ | 120M | $\mathbf{0.76 \pm 0.01}$ | 120M | $0.60 \pm 0.00$ | 120M |
| Tabby MH GPT2 | $\mathbf{0.87 \pm 0.01}$ | 360M | $0.73 \pm 0.03$ | 430M | $0.53 \pm 0.11$ | 430M |
| Base Pythia 160M | $\mathbf{0.87 \pm 0.01}$ | 160M | $\mathbf{0.75 \pm 0.04}$ | 160M | $0.52 \pm 0.11$ | 160M |
| Tabby MH Pythia 160M | $0.86 \pm 0.00$ | 390M | $0.73 \pm 0.02$ | 470M | $0.54 \pm 0.02$ | 470M |
| Base Pythia 410M | $0.86 \pm 0.02$ | 410M | $\mathbf{0.74 \pm 0.03}$ | 410M | $0.28 \pm 0.40$ | 410M |
| Tabby MH Pythia 410M | $\mathbf{0.88 \pm 0.03}$ | 710M | $0.72 \pm 0.05$ | 820M | $0.54 \pm 0.02$ | 820M |
| Base Llama 3.2 1B | $0.82 \pm 0.01$ | 1.2B | $0.73 \pm 0.01$ | 1.2B | $0.29 \pm 0.01$ | 1.2B |
| Tabby MH Llama 3.2 1B | $0.84 \pm 0.02$ | 2.8B | $0.68 \pm 0.09$ | 3.3B | $0.18 \pm 0.26$ | 3.3B |
| Base Llama 3.1 8B | $0.84 \pm 0.01$ | 8.0B | $\mathbf{0.75 \pm 0.01}$ | 8.0B | $0.35 \pm 0.01$ | 8.0B |
| Tabby MH Llama 3.1 8B | $0.86 \pm 0.03$ | 11B | $0.72 \pm 0.01$ | 12B | $0.30 \pm 0.01$ | 12B |

Table 19: Results using GReaT and Tabula training for all three datasets of the experiment in Section 3.2, which compares non-Tabby and Tabby MH models across base LLMs of varying sizes.

|  | Travel | | Diabetes | | House | |
|  | MLE (↑) | Params | MLE (↑) | Params | MLE (↑) | Params |
|---|---|---|---|---|---|---|
| **Non-Synthetic (Upper Bound)** | **0.87** | | **0.73** | | **0.61** | |
| Base Pythia 14m | $0.81 \pm 0.00$ | 14M | $0.60 \pm 0.04$ | 14M | $0.46 \pm 0.06$ | 14M |
| Tabby MH Pythia 14m | $0.81 \pm 0.00$ | 53M | $0.67 \pm 0.01$ | 66M | $0.51 \pm 0.03$ | 66M |
| Base Distilled-GPT2 | $0.86 \pm 0.00$ | 82M | $0.62 \pm 0.00$ | 82M | $0.57 \pm 0.00$ | 82M |
| Tabby MH Distilled-GPT2 | $0.84 \pm 0.00$ | 310M | $0.70 \pm 0.06$ | 390M | $0.56 \pm 0.01$ | 390M |
| Base GPT2 | $0.85 \pm 0.02$ | 120M | $0.64 \pm 0.02$ | 120M | $0.55 \pm 0.00$ | 120M |
| Tabby MH GPT2 | $\mathbf{0.87 \pm 0.03}$ | 360M | $\mathbf{0.74 \pm 0.03}$ | 430M | $0.58 \pm 0.01$ | 430M |
| Base Pythia 160M | $0.81 \pm 0.01$ | 160M | $0.70 \pm 0.01$ | 160M | $0.00 \pm 0.00$ | 160M |
| Tabby MH Pythia 160M | $0.82 \pm 0.02$ | 390M | $0.73 \pm 0.03$ | 470M | $0.54 \pm 0.02$ | 470M |
| Base Pythia 410M | $0.85 \pm 0.01$ | 410M | $0.73 \pm 0.03$ | 410M | $0.53 \pm 0.02$ | 410M |
| Tabby MH Pythia 410M | $0.83 \pm 0.01$ | 710M | $\mathbf{0.74 \pm 0.04}$ | 820M | $0.58 \pm 0.01$ | 820M |
| Base Llama 3.2 1B | $0.82 \pm 0.01$ | 1.2B | $0.70 \pm 0.08$ | 1.2B | $0.53 \pm 0.01$ | 1.2B |
| Tabby MH Llama 3.2 1B | $0.78 \pm 0.03$ | 2.8B | $0.71 \pm 0.03$ | 3.3B | $0.43 \pm 0.08$ | 3.3B |
| Base Llama 3.1 8B | $0.78 \pm 0.04$ | 8.0B | $0.67 \pm 0.01$ | 8.0B | $0.53 \pm 0.01$ | 8.0B |
| Tabby MH Llama 3.1 8B | $0.83 \pm 0.03$ | 11B | $0.73 \pm 0.02$ | 12B | $0.45 \pm 0.00$ | 12B |

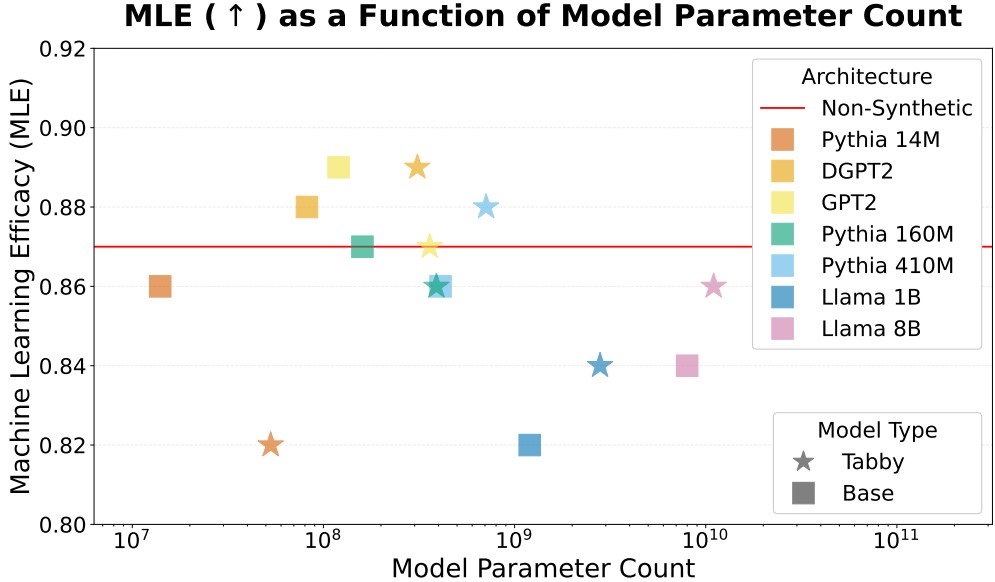

Figure 7: Machine Learning Efficacy (MLE) as a function of parameter count for 7 base LLMs, using Non-Tabby or Tabby MH architectures. Non-Tabby points displayed in blue; MH points in purple. Red line represents Non-Synthetic, upper-bound performance.

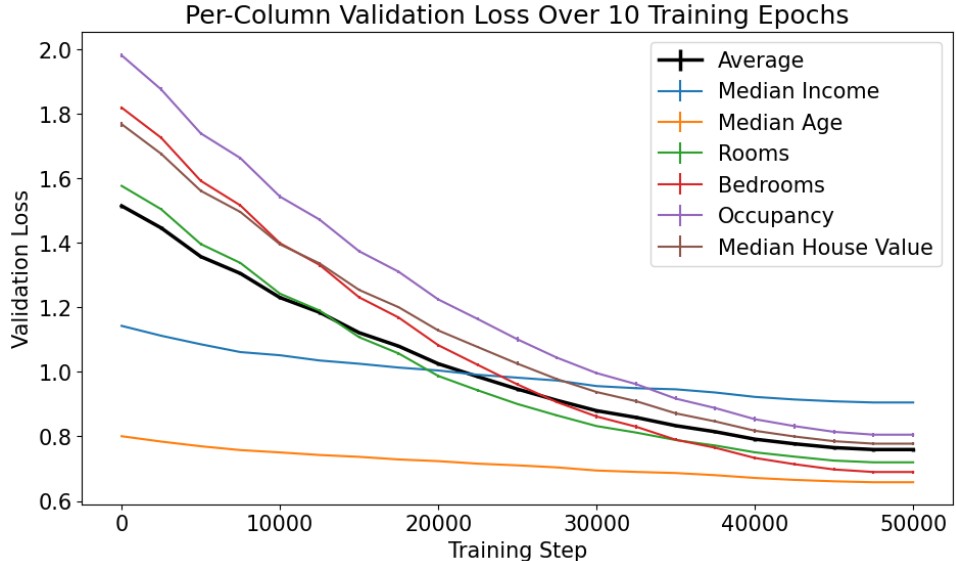

Figure 8: Per-column validation loss across 10 epochs of training Tabby MH Distilled-GPT2 on a subset of House, with average validation loss (black line). While the Occupancy column initially displays the highest loss, Median Income improves little throughout training and becomes the highest-loss column by step 32000.

### E.4 Analysis from Tracking the Adaptation to Individual Columns

Individual column losses are shown in Figure 8. We observe that Occupancy is the largest contributor to the model's loss until step 32000. While Median Income's loss is initially the second-lowest, it improves little throughout the training process and exhibits the highest loss of all columns at the end of training. Additionally, convergence occurs across most columns around step 40000.

Table 20: Five example rows of the synthetic training data used to investigate Claim 5 in Section E.5.

| Code | ID |
|---|---|
| upi_400727_2001 | 427.424.604-d |
| sku_305434_2021 | 60.272.433-i |
| sku_871297_2012 | 727.884.985-G |
| upi_712558_2003 | 160.401.237-E |
| sku_911473_2008 | 856.31.239-c |

These insights are useful in cases where the model struggles to learn some columns more than others. Such information may indicate a need for better preprocessing for a difficult column, or gathering more datapoints to demonstrate the column's distribution. Additionally, the ability to track each column's loss individually and to determine that the losses are roughly balanced across columns, rather than very low in some columns and very high in others, may improve trust in the model—we can understand that there is a low, aleatoric error in each column as opposed to a sizeable epistemic error in a few columns.

### E.5 Investigating Tabby's Generalization to Unseen and High-Cardinality Categorical Features

We now seek to demonstrate our fifth claim.
**Claim 5:** Tabby's language modeling capabilities enable it to capture the underlying semantic structure of column values unseen during pretraining, allowing it to generate novel yet realistic values beyond the pretraining distribution.

**Setup:** We train a Llama 3-8B Tabby MH model for one epoch (LR $= 1e-4$) on a 5000-row synthetic dataset, designed to imitate a common business scenario such as a product database. The dataset includes two columns: "Code" contains a 6-digit random number, prepended by "upi_" or "sku_" and followed by a year [2000,2026]. "ID" contains three numbers, where the first number is [1,999] and other numbers are [0,999], followed by a hyphen and a single lowercase or uppercase letter. We ensure that each trainset Code and each ID are unique. Some examples are in Table 20.

**Results:** We find that the model produces examples of the same structure as the training data. In particular, out of 1000 generated samples:

- 1000/1000 rows meet all of the rules for the Code column value,

- 999/1000 rows meet all of the rules for the ID column value (one row omitted the final hyphen and letter),

- Only one Code value in the trainset reappears in the synthetic set,

- Only one synthetic Code value is reused across the synthetic set,

- All synthetic IDs are unique and

- No trainset IDs appear in the synthetic set.

**Discussion:** Tabby's generalization capabilities are particularly valuable in real-world settings where datasets include non-categorical string features such as names, product IDs, addresses, or telephone numbers. While these capabilities extend Tabby's applicability across diverse data domains, the model's behavior in such cases depends on the implementation details of its underlying language model and tokenizer.

For example, when out-of-vocabulary (OOV) tokens appear, their handling is determined by the specific tokenizer used. In the Distilled-GPT2 backbone employed in our main experiments, the tokenizer applies a byte-level variant of Byte Pair Encoding (BPE) (Radford et al., 2019), following the subword tokenization approach of Sennrich et al. (2016). This ensures that every possible UTF-8 string can be represented without resorting to dedicated OOV tokens. Instead, previously unseen strings are decomposed into byte-level subtokens, which the model can interpret and adapt to based on surrounding context. By contrast, other tokenizers may rely on explicit OOV symbols or restricted vocabularies, potentially affecting Tabby's ability to generalize to novel strings and impacting overall performance in such settings.

Table 21: Learning rates for LLM results presented in Section 3.1.

|  | Diabetes | Travel | Adult | Magic | Shoppers | Abalone | Rainfall | House |
|---|---|---|---|---|---|---|---|---|
| *Plain* Non-Tabby | $1e-4$ | $1e-4$ | $1e-4$ | $1e-4$ | $1e-4$ | $1e-4$ | $1e-4$ | $1e-4$ |
| *Plain Tabby MMLP* | $1e-4$ | $1e-4$ | $1e-4$ | $1e-4$ | $1e-4$ | $1e-4$ | $1e-4$ | $1e-4$ |
| *Plain Tabby MH* | $1e-4$ | $1e-4$ | $1e-4$ | $1e-4$ | $1e-4$ | $1e-4$ | $1e-4$ | $1e-4$ |
| *Plain Tabby MMLP+MH* | $1e-4$ | $1e-4$ | $1e-4$ | $1e-4$ | $1e-4$ | $1e-4$ | $1e-4$ | $1e-4$ |
| GReaT Non-Tabby | $1e-4$ | $1e-4$ | $1e-4$ | $1e-4$ | $1e-4$ | $1e-4$ | $1e-4$ | $1e-4$ |
| GReaT *Tabby MMLP* | $1e-4$ | $1e-4$ | $1e-4$ | $1e-4$ | $1e-4$ | $1e-4$ | $1e-4$ | $1e-4$ |
| GReaT *Tabby MH* | $1e-6$ | $1e-4$ | $1e-4$ | $1e-4$ | $1e-4$ | $1e-4$ | $1e-4$ | $1e-4$ |
| GReaT *Tabby MMLP+MH* | $1e-4$ | $1e-4$ | $1e-4$ | $1e-4$ | $1e-4$ | $1e-4$ | $1e-4$ | $1e-4$ |
| GTT Non-Tabby | $1e-4$ | $1e-4$ | $1e-4$ | $1e-4$ | $1e-4$ | $1e-4$ | $1e-4$ | $1e-4$ |
| GTT *Tabby MMLP* | $1e-4$ | $1e-4$ | $1e-4$ | $1e-4$ | $1e-4$ | $1e-4$ | $1e-4$ | $1e-4$ |
| GTT *Tabby MH* | $1e-6$ | $1e-4$ | $1e-4$ | $1e-4$ | $1e-4$ | $1e-4$ | $1e-4$ | $1e-4$ |
| GTT *Tabby MMLP+MH* | $1e-4$ | $1e-4$ | $1e-4$ | $1e-4$ | $1e-4$ | $1e-4$ | $1e-4$ | $1e-4$ |

Table 22: Learning rates for *Plain-trained* LLMs of varying sizes in Section 3.2.

| *Plain Training* | | | |
|---|---|---|---|
|  | Travel | Diabetes | House |
| Base Pythia 14M | $1e-4$ | $1e-4$ | $1e-4$ |
| *Tabby MH* Pythia 14M | $1e-6$ | $1e-4$ | $1e-4$ |
| Base Distilled-GPT2 | $1e-4$ | $1e-4$ | $1e-4$ |
| *Tabby MH* Distilled-GPT2 | $1e-4$ | $1e-4$ | $1e-4$ |
| Base GPT2 | $1e-4$ | $1e-4$ | $1e-4$ |
| *Tabby MH* GPT2 | $1e-4$ | $1e-4$ | $1e-4$ |
| Base Pythia 160M | $1e-4$ | $1e-4$ | $1e-6$ |
| *Tabby MH* Pythia 160M | $1e-4$ | $1e-4$ | $1e-4$ |
| Base Pythia 410M | $1e-6$ | $1e-4$ | $1e-6$ |
| *Tabby MH* Pythia 410M | $1e-6$ | $1e-6$ | $1e-4$ |
| Base Llama 3.2 1B | $1e-6$ | $1e-4$ | $1e-6$ |
| *Tabby MH* Llama 3.2 1B | $1e-6$ | $1e-4$ | $1e-6$ |
| Base Llama 3.1 8B | $1e-6$ | $1e-6$ | $1e-6$ |
| *Tabby MH* Llama 3.1 8B | $1e-6$ | $1e-6$ | $1e-6$ |

### E.6 Hyperparameters for All Experiments

We list the learning rates chosen for Section 3.1 in Table 21, Section 3.2 in Table 22 and Section 24 in Table 24. We select the learning rate that yields lowest training loss from the set $\{1e-3, 1e-4, 1e-6, 1e-8\}$. For non-LLM methods in our experiments, we use the hyperparameters recommended by their respective papers.

Table 23: Learning rates for *GReaT (plus TapTap)-trained* LLMs of varying sizes in Section 3.2.

### GReaT + TapTap Training

|  | Travel | Diabetes | House |
|---|---|---|---|
| Base Pythia 14M | $1e-4$ | $1e-6$ | $1e-4$ |
| *Tabby MH* Pythia 14M | $1e-6$ | $1e-6$ | $1e-4$ |
| Base Distilled-GPT2 | $1e-4$ | $1e-4$ | $1e-4$ |
| *Tabby MH* Distilled-GPT2 | $1e-4$ | $1e-4$ | $1e-4$ |
| Base GPT2 | $1e-4$ | $1e-4$ | $1e-4$ |
| *Tabby MH* GPT2 | $1e-4$ | $1e-4$ | $1e-4$ |
| Base Pythia 160M | $1e-4$ | $1e-6$ | $1e-4$ |
| *Tabby MH* Pythia 160M | $1e-6$ | $1e-6$ | $1e-6$ |
| Base Pythia 410M | $1e-6$ | $1e-6$ | $1e-6$ |
| *Tabby MH* Pythia 410M | $1e-4$ | $1e-6$ | $1e-6$ |
| Base Llama 3.2 1B | $1e-4$ | $1e-4$ | $1e-6$ |
| *Tabby MH* Llama 3.2 1B | $1e-4$ | $1e-4$ | $1e-4$ |
| Base Llama 3.1 8B | $1e-4$ | $1e-4$ | $1e-6$ |
| *Tabby MH* Llama 3.1 8B | $1e-4$ | $1e-4$ | $1e-6$ |

Table 24: Learning rates for JSON Glaucoma (Manoj, 2024) experiment presented in Section 3.4.

|  | Glaucoma |
|---|---|
| Base DGPT2 | $1e-4$ |
| *Tabby MH* DGPT2 | $1e-4$ |

