# OpenReview forum: "Tabby: A Language Model Architecture for Tabular and Structured Data Synthesis"
_TMLR — Accepted by TMLR_

### Review · Reviewer_XGrn · 2025-10-02

**Summary Of Contributions:**

Tabby is a architectural modification to transformers to enable more effective modeling of tabular data by assigning a small set of parameters to modeling each field in the table or other structured data (i.e., by designating N heads in the final layer, each responsible for generating the content of one column). This, coupled with a technique for formatting tabular data for training called Plain, enables much better learning of tabular datasets and realistic synthetic data generation in-domain.

**Audience:**

Yes

**Audience Explanation:**

While I don't work in this area, so I cannot say for sure, the work motivates itself well in the domain of tabular data modeling and appears to engage deeply with the relevant literature, in addition to presenting improvements over existing methods. I also think the idea of a method that enables monitoring loss per-column is quite nice, and I could see this being useful.

**Broader Impact Concerns:**

The broader impacts are sufficiently addressed.

**Claims And Evidence:**

Yes

**Claims Explanation:**

The submission's key claims are that the architectural modification Tabby and data processing modification Plain improve tabular modeling. The paper presents results for these both individually and together across several reasonable datasets, and using a number of metrics for tabular data learning.

**Requested Changes:**

Clarifications:
* While I don't think it's incorrect to describe this as an MoE, the paper is not always clear on the detail of routing to individual heads (or attention/MLP blocks, for Appendix E). It's my understanding that the tokens corresponding to the $i$th column are always routed to the $i$th block-- is this correct? If so, this could be clarified in the text. The general reader's understanding of an MoE will be a model with a learned router, possibly one aiming to load balance across experts.
* Could you clarify in the training section: you state that the loss for each column can be computed separately, which makes sense for the differentiated heads. Do you backprop between columns, or at the end of a batch of rows? I imagine it's the latter, but clarification would be useful.
* Why does Tabby add so many parameters in Table 3? This makes me concerned that I've misunderstood something about the work. Why does adding Tabby for Travel triple the parameter count of GPT-2?

Experimental requests:
* I'm curious about adapting across table formats. The presumed advantage of the prior methods is that, once you have learned something about tabular data, it would be easier to train on a new tabular task. (1) Is this true? And (2) if so, can you provide results showing whether Tabby makes it more difficult to perform this transfer, since it requires architectural adaptation to each dataset? I think the paper should contain either this result or some discussion of tradeoffs here.
* I'm not really convinced by the argument about parameter sharing in 2.4. _Does_ Tabby work well with parameter sharing, and how much does this limit the scaling up of cost with table size? I don't think the argument that this can scale to extremely large tables is necessary for the work, so I would also be okay with just modifying claims in this section to clarify that this is a direction for future work; however, as written, I think the claim needs experimental support.

Optional, but might strengthen the work:
* It's a bit hard to see from Figure 4 why Tabby is better than GTT; a different visualization might help.
* It's very interesting to me that GPT-2 and related models outperform modern 1-8B LMs on this task. Some discussion of this or pointers to discussions of this in tabular modeling would be nice.

---

### Review · Reviewer_1YRv · 2025-10-08

**Summary Of Contributions:**

This paper introduces Tabby, a post-training architectural modification to transformer-based language models that enables higher-fidelity tabular and structured data synthesis through column-specific Mixture-of-Experts (MoE) layers. The work pairs this architecture with Plain, a simplified training technique that outperforms existing LLM-based approaches (GReaT, Tabula) while being computationally more efficient. The proposed approach represents the first application of MoE specifically to LLM-based table generation, and achieves strong empirical results on multiple tested datasets despite being more straightforward and parameter-efficient than baseline approaches. Beyond tabular data, authors also highlight strong performances on other structured data such as hierarchical JSONs, suggesting more general capabilities across structured data modalities.

The main strengths of this work are:

- The architectural innovation of using MoEs for modeling columns in tabular data, stemming from the intuition that columns are distinct-but-interdependent features that benefit from specialized modeling, is sound and leads to considerable gains in downstream performances.

- The evaluation is quite comprehensive, spanning 9 synthesis methods across 6 datasets.

- The use of a per-column loss simplifies the analysis of models during training to identify challenging data that require further processing, as illustrated in the example of Appendix E.4.

- The simple Plain approach also makes the method easier to adopt and less dependent on an adequate choice of additional procedures, e.g. tabular pretraining for TapTap.

- Authors provide detailed hyperparameter evaluations in Table 17-20 to ensure the reported results are reproducible.

The weaknesses of this work are the following:

- The inclusion of more modern approaches as baselines, e.g. HARMONIC [1] and CLLM [2] for LLM-based methods or CTAB-GAN+ [3] would have provided additional evidence that the proposed method obtains strong performances compared to recent work in this domain.

- The problematic results of GReaT for the Rainfall dataset in Table 2 are not justified in light of practical improvement. It would have been helpful to clarify what could be the cause of such failures, and why the Tabby approach resolves them.

[1] Wang et al. 2024. HARMONIC: Harnessing LLMs for Tabular Data Synthesis and Privacy Protection. https://arxiv.org/abs/2408.02927v1
[2] Seedat et al. 2024. Curated LLM: Synergy of LLMs and Data Curation for tabular augmentation in low-data regimes. https://arxiv.org/abs/2312.12112
[3] Zhao et al. 2023. CTAB-GAN+: Enhancing Tabular Data Synthesis. https://arxiv.org/abs/2204.00401

**Audience:**

Yes

**Audience Explanation:**

This paper addresses a critical gap at the intersection of foundation models and structured data, a topic of growing importance as the field recognizes the ubiquity of tabular data, despite it receiving far less attention than text/images. Several factors make this work highly relevant to TMLR's audience:

- Van Breugel & van der Schaar's ICML 2024 position paper "Why Tabular Foundation Models Should Be a Research Priority" explicitly called for architectural innovations enabling cross-dataset generalization for tables. Tabby directly responds to this call with the first architecture modification (vs. pure fine-tuning approaches) designed explicitly for tabular generation.

- Tabular data dominates real-world ML applications—healthcare records, financial transactions, scientific experiments, business analytics. Improved synthesis methods impact privacy-preserving data sharing (healthcare, finance), with the potential to enable data augmentation through the safe sharing of synthetic versions of proprietary datasets. The variety of datasets tested in this work and the extension to JSON establishes its generality beyond narrow application domains.

- The first application of MoEs for tabular data opens potential avenues for further improvements in this area, e.g. an feature-type-specific experts (numerical vs. categorical) improve performance? From an interpretability perspective, expert specialization might improve transparency by enabling feature-importance analysis through expert weights.

Finally, the effectiveness of the presented Plain method challenges conventional wisdom from previous approaches, such as Tabula and GReaT, requiring additional processing steps. The fact that small Tabby models exceed large baseline LLMs contradicts the field's general scaling trend, suggesting architectural specialization provides greater returns than brute-force parameter increases for structured domains.

**Broader Impact Concerns:**

The paper includes a "Broader Impact Statement" (Section 5) acknowledging dual-use concerns. The current statement provide a sufficient, if slightly generic, overview of the current risks associated with synthetic tabular data generation.

**Claims And Evidence:**

Yes

**Claims Explanation:**

The paper's core architectural claim, i.e. that MoE modifications improve LLM-based tabular synthesis, has some empirical support across various datasets and model scales. Table 2 shows that Plain Tabby MH achieves the best performance on 4 out of 6 datasets, although the Tab-DDPM method is within range for two of those datasets. The extensibility claim for JSON data (Table 4) is particularly interesting, with Tabby being the only method that reaches the non-synthetic upper-bound MLE (0.97).

The first claim, "Plain-trained Tabby models generate higher-quality tabular data than prior approaches," relies on comparison with methods that are not very recent (e.g. GReaT from 2022). The inclusion of methods like the ones mentioned above might have provided more convincing evidence of the usefulness of this adaptation in the context of tabular data synthesis.

The second claim, "Tabby allows smaller LLMs to achieve similar or better synthetic data fidelity than LLMs with higher parameter counts", is confirmed by results in Table 3, but the small margins between non-Tabby and Tabby methods puts in question whether Tabby proves useful in this setting.

The two remaining claims are adequately supported by evidence provided by the JSON evaluation in Section 3.3, and the analysis of loss curves on the House dataset in Section 3.4.

**Requested Changes:**

- The inclusion of additional baseline comparisons with more recent methods in Section 3.1 would confirm the validity of the first claim made in this paper. In particular, beyond [1], [2] and [3], authors could consider also [4] and [5], which represent current diffusion SOTA for tabular data synthesis. If these methods cannot be run, provide detailed explanation of why (e.g., code unavailability, computational constraints) and acknowledge this limitation explicitly.

- Regarding the privacy claims in Section E.1, while common Discrimination and DCR metrics were provided (Tables 6-7),  these were recently shown by [6] to be inadequate for identifying privacy leakages. In light of this, claims about privacy should be supported with a proper MIA evaluation.

- Justify "efficient" and "lightweight" claims with quantitative evidence compared to baselines, not only in terms of parameter count but also training and inference time. Various other methods provide training time references (e.g. 9+ hours for GReaT, Table 6), and those would also be helpful here to conduct a fairer evaluation of different methods' efficiency.

- Some additional clarity regarding the implementation would be useful. In particular, for Tabby MH: which specific layer(s) are replaced with MoE? (only final output layer, or multiple heads?). A mathematical characterization of the gating function beyond the "Gated Mixture-of-Experts" mention would also add clarity. Finally, regarding initialization: "weights for each block in Λa are initialized to equal the weights of La" (Section 2.3), does this mean all experts start identical then diverge during training?



[4] Shi et al. 2024. TabDiff: a Mixed-type Diffusion Model for Tabular Data Generation. https://arxiv.org/abs/2410.20626
[5] Zhang et al. 2024. Mixed-Type Tabular Data Synthesis with Score-based Diffusion in Latent Space. https://arxiv.org/abs/2310.09656
[6] Yao et al. 2025. The DCR Delusion: Measuring the Privacy Risk of Synthetic Data. https://arxiv.org/abs/2505.01524

---

### Review · Reviewer_i7HK · 2025-10-30

**Summary Of Contributions:**

The paper proposes Tabby, a post‑training architectural modification for transformer LLMs that replaces selected blocks with Mixture‑of‑Experts (MoE) so that each tabular column (or structured feature) is modeled by its own expert.  The best‑performing variant Tabby Multi‑Head swaps the LM head for a per‑column MoE head. The paper also introduces Plain, a simple finetuning recipe that serializes each row as text with an end-of-column token `<EOC>`, enabling per-column loss tracking and straightforward parsing at inference. Together, Tabby+Plain target higher-fidelity tabular (and nested JSON) synthesis using pretrained LLMs rather than training tabular models from scratch. The authors claim gains in machine learning efficacy (MLE) across six datasets.

**Audience:**

Yes

**Audience Explanation:**

This paper introduces Tabby, a per-column MoE add-on for pretrained LLMs, paired with a simple ``Plain'' row-as-text training scheme, and demonstrates clear, practical advances as follows:


1. **Targeted architectural idea (per-column MoE) with simple and clear motivation.**
The per-column expert heads increase expressivity for heterogeneous feature types while still allowing cross-column dependencies, and the Multi-Head variant is especially effective. The change is conceptually simple and can be added to existing pretrained language models without needing to retrain them completely.

2. **Plain training is practical and transparent.**
Representing each row as a linearized text sequence with explicit `<EOC>` token keeps fine-tuning and inference simple.
Despite its simplicity, **Plain** consistently matches or outperforms some more complex LLM-based tabular training methods in the experiments.

3. **Extends beyond flat tables to structured (nested) data.**
The recursive application to JSON attains parity with real data and outperforms non‑LLM and diffusion baselines in that setting, supporting the generality claim.

**Broader Impact Concerns:**

There is no broader impact concerns.

**Claims And Evidence:**

Yes

**Claims Explanation:**

The submission's claims are convincingly supported by quantitative evidence: benchmark tables show Plain-trained Tabby-MH achieves the best or near-best MLE on most studied datasets, even reaching the non-synthetic upper bound on Diabetes, Travel, and Adult.  Aggregate performance-profile curves further give Tabby-MH the highest AUP across methods, indicating superior overall performance.  And on a nested-JSON glaucoma dataset, Tabby-MH matches real-data MLE while attaining the lowest discrimination score, outperforming baselines.

**Requested Changes:**

## Weaknesses / Limitations

1. **Coverage of strongest recent diffusion methods**
The comparison includes some diffusion-based baselines (Tab-DDPM, Forest Diffusion) but does not cover the most recent SoAT diffusion models (e.g., Tabsyn, TabDiff). Consequently, it remains difficult to pinpoint Tabby’s position relative to the newest diffusion methods. Notably, TabDiff leverages the **Diffusion LLM** concept for tabular data synthesis, which is conceptually related to this paper's approach.

[Tabsyn] Zhang et al. -- Mixed-Type Tabular Data Synthesis with Score-based Diffusion in Latent Space,  ICLR 2024

[TabDiff] Shi et al. -- TabDiff: a Mixed-type Diffusion Model for Tabular Data Generation, ICLR 2025

2. **Extremely high-cardinality categorical fields.**
The approach relies on tokenizing raw column values as text, but columns with millions of distinct IDs (mostly unseen in pretraining) are not explicitly studied. It is unclear how Tabby behaves when most category values are effectively OOV strings.


## Suggestions for Improvement

1. **Include recent diffusion baselines**
Building on the Weaknesses section, please add **TabSyn** and **TabDiff** as baselines to clarify how **Tabby** and **Plain** compare with recent tabular diffusion models, and evaluate Tabby/Plain on the datasets used in those papers (at least two beyond **Adult**, which you already cover).


2. **Expand on handling high-cardinality categorical features**
Discuss how Tabby (and Plain) behave with columns that have extremely high cardinality or free-form IDs, and whether additional preprocessing (e.g., hashing, learned embeddings) is needed at scale.

---

> ### Author Response · Authors · 2025-11-04
>
> We wish to thank Reviewer i7HK for their nuanced feedback on Tabby, as well as their appreciation for Tabby’s simplicity and practical performance improvements compared to prior works. The reviewer highlighted two areas for improvement, which we address as follows:
>
> **Recent (Diffusion) Baselines and Their Datasets**
>
> *Baselines:* We agree on the importance of including the two recent diffusion-based baselines of TabSyn [1] and TabDiff [2]. Beyond these two, we have also added results for three additional recent approaches: LLM-based CLLM [3] and HARMONIC [4], along with GAN-based CTAB-GAN+ [5]. Please see the MLE results for each of these in the table below, which indicate that **Tabby achieves best performance in 5/6 datasets**. Best performance is defined as higher than non-synthetic performance, or the highest-performing synthesis method overall (details in Section 3.0 of the manuscript).
>
> | **Name**             | **Diabetes**  | **Travel**    | **Adult**     | **Abalone**   | **Rainfall**  | **House**     |
> |----------------------|---------------|---------------|---------------|---------------|---------------|---------------|
> | Non-Synthetic        | **0.73**      | **0.87**      | **0.85**      | **0.45**      | **0.54**      | **0.61**      |
> | CTAB-GAN+            | 0.62±0.00     | 0.81±0.00     | 0.76±0.00     | 0.24±0.03     | 0.22±0.15     | 0.55±0.00     |
> | HARMONIC             | 0.64          | 0.83          | 0.76          | 0.32          | 0.08          | 0.30          |
> | CLLM                 | **0.74±0.02** | 0.83±0.03     | 0.80±0.02     | 0.00±0.00     | 0.00±0.00     | N/A           |
> | TabSyn               | 0.65±0.01     | 0.74±0.13     | 0.80±0.04     | 0.13±0.23     | 0.45±0.00     | 0.60±0.01     |
> | TabDiff              | **0.75±0.02** | 0.86±0.02     | 0.83±0.01     | 0.41±0.01     | 0.43±0.02     | **0.61±0.00** |
> | **Plain Tabby MH DGPT2** | **0.74±0.00**  | **0.88±0.01** | **0.85±0.00** | **0.43±0.01** | **0.49±0.00** | 0.60±0.00     |
>
> *Datasets:* While our submission shares two datasets with TabDiff (Adult and Diabetes), we additionally evaluate Tabby alongside TabSyn and TabDiff on three datasets that appear in both the TabSyn and TabDiff papers. Details for all datasets are provided in Table 6 of [2]. For the new datasets (Shoppers, Magic and Beijing) we conduct a single run following the methodology used in our main experiments, using a learning rate of 1e-4 for Tabby, the recommended hyperparameters for TabSyn/TabDiff, and sampling 10,000 datapoints. As shown by the MLE results in the table below, **Tabby achieves the highest performance on 4/5 datasets.**
>
> | **Name**             | **Shoppers** | **Magic**    | **Beijing**  | **Diabetes**      | **Adult**         |
> |----------------------|----------|----------|----------|---------------|---------------|
> | Non-Synthetic        | **0.88** | **0.82** | **0.27** | **0.73**      | **0.85**      |
> | TabSyn               | **0.88** | **0.82** | 0.25     | 0.65±0.01     | 0.80±0.04     |
> | TabDiff              | **0.89** | 0.81     | 0.26     | **0.75±0.02** | 0.83±0.01     |
> | **Plain Tabby MH DGPT2** | **0.88** | 0.80     | **0.29** | **0.74±0.00** | **0.85±0.00** |

---

> > ### Author Response · Authors · 2025-11-04
> >
> > **High-Cardinality and OOV Categorical Features**
> >
> > *High-Cardinality Features:* Even when a given column is high-cardinality, with values unseen during pretraining, Tabby is still capable of picking up on any semantic structure contained within the column. To demonstrate this ability, we create a synthetic 5000-row dataset with two columns, designed to represent a common enterprise scenario such as a product database. The Code column contains a 6-digit random number, prepended by “upi_” or “sku_” and followed by a year [2000,2026). The ID column contains three numbers, where the first number is [1,999] and other numbers are [0,999], followed by a hyphen and a single lowercase or uppercase letter. We ensure that each Code and each ID are unique. Some examples are in the table below.
> >
> > | Code                  | ID                  |
> > |-----------------------|---------------------|
> > | ```upi_400727_2001``` | ```427.424.604-d``` |
> > | ```sku_305434_2021``` | ```60.272.433-i```  |
> > | ```sku_871297_2012``` | ```727.884.985-G``` |
> > | ```upi_712558_2003``` | ```160.401.237-E``` |
> > | ```sku_911473_2008``` | ```856.31.239-c```  |
> >
> > After training a Llama 3 8B Tabby model for just one epoch (LR of 1e-4), the model produces examples of the same structure as the training data. In particular, out of 1000 generated samples:
> > - 1000/1000 rows meet all of the rules for the Code column value,
> > - 999/1000 rows meet all of the rules for the ID column value (one row omitted the final hyphen and letter),
> > - Only one Code value in the trainset reappears in the synthetic set,
> > - Only one synthetic Code value is reused across the synthetic set,
> > - All synthetic IDs are unique and
> > - No trainset IDs appear in the synthetic set.
> >
> > **The ability to learn the structure of column values, as opposed to merely memorizing a closed set of values occurring in the training data, is a strength of Tabby compared to other approaches.** The diffusion- and GAN-based approaches evaluated in our work are not capable of parsing the internal structure of categorical columns, while tabular encoding techniques for LLMs such as Tabula [6] hide this internal structure from the LLM entirely.
> >
> > There are also additional cases in which the dataset contains tokens that are not recognized by the LLM or its tokenizer at all. Here, the behavior of Tabby is determined by the behavior of its underlying LLM. For the Distilled-GPT2 backbone used in our main experiments, the tokenizer employs a byte-level variant of Byte Pair Encoding (BPE) [7], building on the subword tokenization approach of [8] to ensure that every possible UTF-8 string can be represented without out-of-vocabulary tokens. Instead of relying on OOV tokens, novel strings are decomposed into byte-level tokens, which the model can interpret and adapt to based on surrounding context.
> >
> > We thank the reviewer again for recommending these additions and improvements to our paper, and would appreciate any additional feedback.
> >
> >
> > [1] Zhang et al. 2024. Mixed-Type Tabular Data Synthesis with Score-based Diffusion in Latent Space.
> > [2] Shi et al. 2024. TabDiff: a Mixed-type Diffusion Model for Tabular Data Generation.
> > [3] Seedat et al. 2024. Curated LLM: Synergy of LLMs and Data Curation for tabular augmentation in low-data regimes.
> > [4] Wang et al. 2024. HARMONIC: Harnessing LLMs for Tabular Data Synthesis and Privacy Protection.
> > [5] Zhao et al. 2023. CTAB-GAN+: Enhancing Tabular Data Synthesis.
> > [6] Zhao et al. 2025. Tabula: Harnessing Language Models for Tabular Data Synthesis.
> > [7] Radford, Alec, et al. “Language Models are Unsupervised Multitask Learners.” OpenAI Technical Report, 2019.
> > [8] Sennrich, Rico, et al. “Neural Machine Translation of Rare Words with Subword Units.” Proceedings of ACL, 2016.

---

### Author Response · Authors · 2025-11-09
**Revised Manuscript Incorporating Reviewer Feedback**

We thank the Action Editor and all reviewers for their thoughtful and constructive feedback. In response, we have revised the manuscript to incorporate the requested clarifications, additional analyses, and new experimental results discussed in our rebuttal. All updates are highlighted in blue throughout the paper. We believe these revisions strengthen the presentation and comprehensively address the reviewers’ concerns. We would be glad to provide any additional information if needed.

---

### Decision · Action_Editor_ow2e · 2025-12-05

**Recommendation:** Accept as is

**Audience:**

Yes

**Audience Explanation:**

All reviewers agree.  It provides a small but clear improvement in an active area.

**Claims And Evidence:**

Yes

**Claims Explanation:**

All reviewers agree with this.  The paper looks very carefully done.